# Screening of *Macadamia integrifolia* Varieties Based on the Comparison of Seedling Adaptability and Quality Differences

**DOI:** 10.3390/biology14121638

**Published:** 2025-11-21

**Authors:** Xibin Zhang, Xu Li, Liangyi Zhao, Zhitao Yang, Chengping Luo, Fuyan Ma, Weifeng Zhao, Baoqiong Zhang, Wenxiu Yang, Xuehu Yang, Liangliang Sun

**Affiliations:** 1College of Tropical Crop, Yunnan Agricultural University, Kunming 650201, China; 18760944756@163.com (X.Z.); lxamhc@163.com (X.L.); 19915889566@163.com (L.Z.); 15508724987@163.com (Z.Y.); 18314184251@163.com (C.L.); 18287814895@163.com (F.M.); zwf9721@163.com (W.Z.); 2010073@ynau.edu.cn (B.Z.); xiaoyangyang111@126.com (W.Y.); 2International Joint Laboratory for Green Production Technology of Special Fruits and Vegetables in Yunnan Province, Kunming 650201, China

**Keywords:** variety selection, seedling adaptability, antioxidant enzymes, nut quality, principal component analysis

## Abstract

Macadamia is a high-value cash crop; however, the varietal differences in seedling adaptability and fruit quality have not been sufficiently studied. This research systematically compared the performance of three macadamia varieties (A4, A16, A203) in terms of seedling growth and fruit quality. The study revealed distinct characteristics among the varieties regarding stress resistance, flavor, and nutritional functions: A4 exhibited stronger stress resistance, making it suitable for cultivation under adverse conditions; A16 demonstrated superior fruit flavor and higher amino acid content, rendering it suitable for fresh consumption and high-value product development; A203 excelled in mineral and medicinal amino acid content, indicating high potential for nutritional and health benefits. We established a comprehensive evaluation system encompassing “morphology—photosynthesis—antioxidant activity—amino acids—quality,” which provides a scientific basis for variety selection, cultivation management, and industrial development of macadamia.

## 1. Introduction

Macadamia (*Macadamia* spp.), an evergreen tree fruit crop belonging to the Proteaceae family and the genus *Macadamia* F. Muell., is native to the subtropical rainforests of southeastern Queensland, Australia [1]. Its kernel is renowned for its rich nutritional profile and unique flavor [2], earning it the title “the queen of nuts” and granting it high economic and scientific value in international markets [3]. As a high-value cash crop, macadamia is increasingly cultivated in tropical and subtropical regions for its premium edible nuts, which command strong market prices and contribute significantly to local economies. In recent years, with the rapid growth of the health food market, macadamia has found increasingly wide applications in the food, health product, and cosmetics industries, demonstrating promising market prospects [4,5]. The composition of high-quality fats, proteins, carbohydrates, and mineral elements in macadamia kernels has been well characterized [6,7,8]. Systematic analyses have also been conducted on the types, contents, and taste characteristics—such as umami, sweetness, and medicinal amino acids—of free amino acids (FAAs) and hydrolyzed amino acids (HAAs), revealing their significant impact on kernel flavor and nutritional quality [9,10]. Therefore, selecting superior varieties with enhanced adaptability and quality traits is crucial for improving production efficiency and industrial competitiveness. Furthermore, bioactive components like phenolics and flavonoids, which are abundant in the husk, have been confirmed to possess health benefits such as blood lipid reduction and anti-cancer properties [11,12]. Variations in nutritional components among different cultivars and growing regions provide a scientific basis for their functional development.

Cultivar selection represents a pivotal strategy for enhancing macadamia production efficiency. Substantial research has evaluated regional adaptability through agronomic traits including tree architecture, yield performance, and leaf anatomical features, providing valuable guidance for geographical cultivation planning [13,14,15]. However, current evaluation systems predominantly focus on mature trees, leaving a considerable knowledge gap regarding systematic comparison of seedling physiological characteristics. Seedling vigor can be comprehensively assessed through multiple foliar indicators. Phenotypic traits such as leaf dimensions and leaf shape index directly reflect growth status, with shorter petioles and reduced leaf shape index typically indicating robust development [16]. Photosynthetic efficiency can be gauged through chlorophyll content, photosynthetic rate (Pn), and stomatal conductance (Cleaf), while antioxidant capacity and stress resilience are reflected in enzymatic activities and malondialdehyde (MDA) levels [17]. Additionally, dynamic changes in free amino acid composition and concentration serve as reliable indicators for nitrogen metabolism intensity and stress adaptation [18,19,20]. Fruit quality assessment encompasses multiple dimensions including nut morphology (longitudinal diameter, mass, kernel recovery rate), mineral composition, amino acid profiles, and husk phenolic content, collectively enabling comprehensive cultivar evaluation [21,22].

Although previous research has conducted regional adaptability evaluations for introduced macadamia cultivars, systematic investigation into physiological differences at the seedling stage—particularly regarding leaf free amino acids and comprehensive multi-indicator assessments—remains limited. More importantly, existing studies often evaluate seedling adaptability and fruit quality in isolation, lacking an integrated framework that links early physiological performance with final product attributes. To address this gap, we established a comprehensive multi-trait evaluation system encompassing “morphology–photosynthesis–antioxidant activity–amino acids–quality,” which simultaneously assesses seedling adaptability and mature fruit quality. Therefore, we utilized cultivars A4, A16, and A203 as plant materials. We measured seedling leaf morphology, photosynthetic parameters, antioxidant enzyme activities, and free amino acid profiles, combined with analyses of kernel nutritional components and husk phenolic substances, to systematically compare seedling adaptability and fruit quality differences among the cultivars. These findings provide theoretical support for elite cultivar selection and industry development.

## 2. Materials and Methods

### 2.1. Plant Materials

Three macadamia cultivars (A4, A16, and A203), characterized by distinct growth traits, widespread cultivation in Yunnan Province, and promising development prospects, were selected for this study based on comprehensive considerations of genetic diversity, economic value, research background, and practical circumstances. Their specific characteristics are as follows: (1) A4: This cultivar exhibits early flowering, spiny leaf margins, and a high degree of self-sterility, with relatively poor compatibility with common rootstocks. It is resistant to husk spot disease, exhibits strong cold and wind tolerance, but is susceptible to high-temperature stress. A4 is suitable for processing and produces high-quality kernels. Its cultivation area in Yunnan Province is expanding. (2) A16: Possessing some hybrid characteristics, A16 grows vigorously with moderately dense branches. It is high-yielding, producing medium-sized, uniform nuts with high kernel recovery, high percentages of first-grade and whole kernels, and excellent kernel quality. Its pollination biology is not well-documented, but it is typically cultivated in mixed orchards to ensure cross-pollination. It demonstrates strong wind, drought, and cold tolerance. Its planting area in Yunnan is also increasing. (3) A203: This cultivar forms a natural shape with natural branching. It flowers and fruits early, bearing nuts in clusters. The fruits are relatively large, and the cultivar is high-yielding with a relatively higher fruit set rate compared to other varieties. It is considered to have a mixed pollination system but benefits from cross-pollination.

For the adaptability experiments, uniformly growing grafted seedlings of these three cultivars were used. For the quality comparative analysis, fresh, green-in-husk nuts newly harvested from these cultivars were utilized. The seedlings of all three cultivars, approximately 1.5 years old, were purchased from Jinghong Jianshun Agricultural Technology Co., Ltd. (Jinghong, China). They were cultivated in the open-field macadamia nursery of the College of Tropical Crops, Yunnan Agricultural University. The nursery site is located in a region with a typical subtropical highland monsoon climate. The seedlings were grown under uniform management practices, including consistent irrigation and fertilization regimes, reflecting common commercial nursery conditions in the region. Mature leaves were collected from these seedlings for physiological indicator measurements related to adaptability. The fresh, green-in-husk nuts were harvested from the Manxieba Nut Base (altitude ~1269 m, 22° N, 100° E) of Pu’er Yunguo Agricultural Technology Co., Ltd. in Simao District, Pu’er City, China. This region is characterized by a subtropical highland climate, with the specific soil and climatic conditions representing a typical production area for macadamia in Yunnan. Nuts were collected from four-year-old plants (fertilized 4–5 times per year) located on sunny slopes at consistent altitudes. Sampling was conducted from the eastern, southern, western, and northern parts of the lower tree canopy. The collected nuts were transported to the laboratory in sealed plastic bags for morphological index measurement. Alternatively, the husks were peeled off, flash-frozen in liquid nitrogen for 8 min, and stored at −80 °C for subsequent use.

### 2.2. Phenotypic Data Collection

Morphological traits were assessed following the method described by Kuan et al. [23]. Descriptive traits such as phyllotaxis, leaf shape, and leaf apex shape were determined via visual observation. Measurable traits, including leaf length, leaf width, petiole length, and leaf shape index, were determined using measurement methods.

The leaf shape index was calculated as Leaf Length/Maximum Leaf Width.

A vernier caliper with an accuracy of 0.02 mm was used to measure the morphological indices of macadamia nuts, as detailed below: (1) Fresh Nut and Nut-in-Shell Horizontal Diameter: Nuts were placed horizontally, and the diameter at the widest point was measured and recorded as the horizontal diameter. (2) Fresh Nut and Nut-in-Shell Vertical Diameter: Nuts were placed vertically, and the diameter at the longest point was measured and recorded as the vertical diameter. (3) Husk Thickness: Three random locations on the nut surface were selected, and the husk thickness was measured using the vernier caliper. The average value was taken as the husk thickness. (4) Weight Measurements: The weights of fresh nuts, fresh nuts-in-shell, fresh husks, and dried husks (constant weight) were measured using an electronic analytical balance (accuracy 0.0001 g, Jinghai, Shanghai, China). For each variety of Australian nut, 30 fruits were measured.

The following indices were calculated:Kernel Recovery Rate (%) = (Fresh Nut-in-Shell Mass/Fresh Nut Mass) × 100Fruit Shape Index = Vertical Diameter/Horizontal DiameterHusk Water Loss = Fresh Husk Weight − Dry Husk Weight (constant weight)

### 2.3. Detection of Leaf Photosynthesis-Related Indicators

The SPAD (Soil and Plant Analyzer Development) value was measured using a SPAD 502 Plus chlorophyll meter (Minolta, Tokyo, Japan). Young leaf tips of consistent maturity, healthy and free from disease, were selected, avoiding veins and edges. The leaf was placed in the measurement clip, ensuring complete coverage of the sensor. Three measurements were taken per leaf and averaged. Thirty leaves were measured per group.

Chlorophyll content was determined entirely under low light conditions. A 0.1 g sample was ground, and 5 mL of 95% ethanol was added until the tissue turned white. The mixture was kept at room temperature for 4 min, then centrifuged at 8000 r/min for 10 min to collect the supernatant. Using 95% ethanol as a blank for zero adjustment, the absorbance was measured at wavelengths of 665 nm, 649 nm, and 470 nm, and the absorbance values were recorded.

Leaf Photosynthetic Parameters: On clear days between 09:00 and 12:00, a photosynthesis system (LI-6800, Li-Cor Inc., Lincoln, NE, USA) was used to measure the net photosynthetic rate (Pn), stomatal conductance (Cleaf), and intercellular CO_2_ concentration (CO_2_int) of mature leaves with consistent maturity.

### 2.4. Detection of Leaf Antioxidant System Physiological Indicators

Malondialdehyde (MDA) content was determined using the thiobarbituric acid (TBA) method. A 0.1 g sample was homogenized in pre-cooled phosphate buffer at low temperature. The supernatant enzyme extract was collected, TBA was added, and the mixture was heated in a boiling water bath for 15 min before being rapidly cooled. After centrifugation at 8000 r/min for 10 min, the supernatant was collected, and its absorbance was measured at 450 nm, 532 nm, and 600 nm.

Superoxide Dismutase (SOD) activity was assayed using the nitroblue tetrazolium (NBT) photoreduction method [24]. Leaf tissue was homogenized with extraction medium, and the supernatant was collected after centrifugation. A 0.1 mL aliquot of the supernatant was mixed with color development reagents. The control group was kept in the dark, while the experimental group was placed in an illuminated incubator at 4000 lux and 28 °C for 15 min. The reaction was terminated by covering with black cloth, and the absorbance was measured at 560 nm.

Peroxidase (POD) and Catalase (CAT) activities were determined according to the method of Aebi and Castillo [25,26]. POD catalyzes the oxidation of guaiacol by H_2_O_2_ to form a tan-colored product. CAT activity was measured based on the consumption of H_2_O_2_ catalyzed by the CAT enzyme.

Superoxide Anion (O^2−^) content was measured using the hydroxylamine oxidation method described by Elstner et al. [27]. O^2−^ oxidizes hydroxylamine to produce nitrite ions (NO^2−^). Under acidic conditions, sulfanilamide reacts with NO^2−^ to form a diazonium salt, which subsequently couples with α-naphthylamine to form a pink azo compound with significant light absorption at 530 nm. A 0.1 g sample was homogenized in phosphate buffer in an ice bath and centrifuged at 4 °C and 1000 r/min for 15 min. Then, 2 mL of the enzyme extract was mixed with 1.5 mL of phosphate buffer and 0.5 mL of hydroxylamine hydrochloride, incubated in a 25 °C water bath for 20 min. Subsequently, 2 mL of this mixture was combined with 2 mL of sulfanilamide and 2 mL of naphthylamine, mixed thoroughly, incubated in a 30 °C constant temperature water bath for 30 min, and the color was measured at 530 nm.

H_2_O_2_ content was determined following the method of Patterson et al. [28]. After establishing a standard curve, a 0.1 g sample was homogenized with a 0.1% TCA solution, and the homogenate was pre-cooled at 4 °C. It was then centrifuged at 4 °C and 1000 r/min for 20 min. One milliliter of the supernatant was mixed with 1 mL of buffer solution and 2 mL of KI, reacted in the dark for 10 min, and the absorbance was measured at 390 nm.

### 2.5. Determination of Free Amino Acid Content

Fresh samples were homogenized with sulfosalicylic acid and refrigerated for 1 h. After centrifugation, the supernatant was collected, passed through a 0.22 μm filter, and then analyzed. An A300 amino acid analyzer (membraPure GmbH, Hennigsdorf, Germany) was employed, utilizing ion-exchange chromatography for amino acid separation. The separated amino acids react with ninhydrin to form colored compounds, which are quantified by a photometer. The conditions were: cation exchange column (length 125 mm, resin particle size 4 μm), column temperature 45–75 °C, reactor temperature 115 °C; buffer flow rate 0.25 mL/min, ninhydrin flow rate 0.125 mL/min; detection wavelengths: 570 nm (Channel A) for 17 amino acids and 440 nm (Channel B) for proline. The injection volume was 20 μL, and the analysis time per sample was approximately 2 h.

### 2.6. Determination of Mineral Element Content

A microwave digestion method using concentrated nitric acid was employed for sample preparation. Kernels were dried to constant weight and ground. The kernel powder was placed into digestion vessels, 7 mL of concentrated nitric acid was added, and the mixture was left to stand overnight. The sealed vessels were then placed in a microwave digestion system. A gradient temperature and pressure program was used: Step 1: 80 °C for 3 min, 10 atm; Step 2: 110 °C for 3 min, 15 atm; Step 3: 140 °C for 3 min, 20 atm; Step 4: 170 °C for 3 min, 25 atm; Step 5: 190 °C for 25 min, 30 atm. After digestion, the vessels were cooled below 80 °C before being opened in a fume hood. After cooling, the vessels were vented until the liquid became clear. The digestate was transferred to a 25 mL volumetric flask. The vessel and lid were rinsed three times with ultrapure water, and the solution was made up to volume. This process was repeated three times. Samples were diluted as necessary. Ultrapure water was used for zero adjustment. An air-acetylene flame atomic absorption spectrophotometer (novAA800F, Jena, Germany) was used. International standard single-element standard solutions (K: GBW(E)080125; Ca: GBW(E)080118; Mg: GBW(E)080126; Fe: GBW 08616; Zn: GBW 08620; Cu: GBW 08615; Mn: GBW(E)080157, National Institute of Metrology, China) were used to prepare and serially dilute standard solutions for K, Ca, Mg, Fe, Mn, Cu, and Zn. Standard curves were constructed based on the concentration gradients and the measured absorbance values, with correlation coefficients (R^2^) between 0.999 and 1.000. Sample concentrations were determined using these standard curves. Quality control was based on standard reference material values and detection limits. The correlation coefficients of the standard curves ranged from 0.9998 to 1.0000. Samples were analyzed, absorbance was recorded, and concentrations were calculated by substituting into the standard curve equations.

### 2.7. Determination of Functional Substance Content

Husks were freeze-dried to constant weight, pulverized using a grinder, and passed through a 200-mesh sieve. A 0.5 g sample of the powder was mixed with 5 mL of 70% methanol, vortexed for 1 min, ultrasonicated for 1 h, vortexed again for 1 min, and left to stand at 4 °C overnight. After centrifugation, the supernatant was collected, filtered through a 0.22 μm membrane into an injection vial. Simultaneously, standard solutions of four phenolics (p-hydroxybenzyl alcohol, 3,4-dihydroxybenzoic acid, p-hydroxybenzoic acid, p-hydroxybenzaldehyde) at different concentrations were prepared using 70% methanol for instrumental analysis. The liquid chromatography conditions were: column, Syncronis C18 (250 × 4.6 mm, 5 μm); mobile phase, 1% acetic acid in water–methanol with gradient elution (gradient table provided); detection wavelength, 260 nm; injection volume, 10 μL; column temperature, 40 °C.

### 2.8. Correlation Analysis

Correlation analysis was performed using the OmicShare Tools platform (https://www.omicshare.com/tools/, accessed on 14 November 2025). Methods appropriate for the data distribution were selected to quantify association strength and direction between variables. The analysis included multi-dimensional indicators: mineral elements, amino acids, fruit phenotypic traits, and phenolic metabolites. Results are presented as correlation heatmaps. Color gradients indicate correlation strength: dark blue (strong negative), white (no correlation), dark red (strong positive). Significance is marked (*, **, ***). Hierarchical clustering dendrograms group variables with similar correlation patterns.

The leaf phenotype vs. physiological indicators heatmap, incorporating hierarchical clustering, covered morphological traits, photosynthesis, antioxidant system, and amino acid metabolism, systematically analyzing their intrinsic links.

The correlation heatmap for inter-group correlations between fruit phenotype and kernel quality included row variables covering over ten mineral elements (e.g., K, Mo, Mg, Mn, Cu, Zn, Fe), various amino acids (e.g., Lys, His, Arg, Ala, Glu, Val), and four phenolic substances (phenol-1, phenol-2, phenol-3, phenol-4). The column variables comprehensively included over ten fruit phenotypic indicators (e.g., fresh nut weight, fresh nut horizontal diameter, fresh nut vertical diameter, fresh nut-in-shell weight, nut-in-shell horizontal diameter, nut-in-shell vertical diameter), indicating correlations between kernel quality indicators and fruit phenotypic indicators.

The intra-group correlation analysis for kernel quality included the same row variables (mineral elements, amino acids, phenolics), reflecting the interrelationships among various kernel quality indicators.

### 2.9. Analysis of Amino Acid Classification

The classification of free amino acids into functional categories (including umami, sweet, bitter, aromatic, medicinal, and essential amino acids) was performed based on established taste and functional attributes reported in the literature [9,10]. Principal component analysis (PCA) was conducted using the OmicShare Tools platform to visualize the overall differences in fruit quality traits among varieties.

### 2.10. Statistical Analysis

Various statistical and bioinformatics tools were employed for data processing and analysis. In preliminary data processing, amino acid content data were analyzed using Aminopeak software (Ver 2.41), and phenolic substance data were preliminarily processed using DA Express software (Ver 2.204.0.661). All data were organized, statistically analyzed, and visualized using WPS Office (W.P.S.20.3109) and SPSS 27 software. Statistical analyses included one-way analysis of variance (ANOVA) and significance tests. The results shown are the means ± SDs (*n* = 3), and different letters indicate significant differences (*p* < 0.05 according to Tukey’s test).

## 3. Results

### 3.1. Apparent Leaf Phenotypic Differences Among Different Macadamia Varieties

Phenotypic evaluation of leaves from macadamia varieties A4, A16, and A203 revealed distinct characteristics. Variety A4 exhibited ternate whorled phyllotaxy in young leaves, featuring ovate leaf blades with acute apices, acuminate bases, and entire margins possessing numerous spines. Young leaves were yellowish-green, maturing to green (Table 1, Figure 1A,B). Variety A16 displayed both ternate and quaternate whorled phyllotaxy. Its young leaves were obovate with obtuse apices, acute bases, and undulate margins bearing few spines. Initially slightly red, they turned light green upon maturation (Table 1, Figure 1A,B). Variety A203 also showed ternate and quaternate phyllotaxy. Young leaves were oblanceolate with sharp-acuminate apices, acuminate bases, and prominently undulate margins containing few spines. They were green when young, deepening to a dark green at maturity (Table 1, Figure 1A,B).

Leaves are the primary sites of photosynthesis, and their growth status reflects overall plant health. Petiole length and leaf shape index can serve as auxiliary indicators of plant vigor [29]. Shorter petioles and a smaller leaf shape index typically signify robust growth, whereas longer petioles and a larger index suggest etiolation. Analysis of mature leaf morphology and petiole length in A4, A16, and A203 seedlings showed that A203 had significantly greater leaf length and width than A4, with A16 intermediate between them (Figure 1C,D). A16 seedlings possessed the shortest petioles. Petiole lengths in A4 and A203 were significantly greater than in A16, by 44.29% and 74.50%, respectively (Figure 1B,E). The leaf shape index was highest in A4, followed by A16 and A203, though differences among the three varieties were not statistically significant (Figure 1F).

### 3.2. Comparison of Leaf Photosynthetic Efficiency Among Different Macadamia Varieties

The SPAD value effectively reflects chlorophyll content per unit leaf area, indirectly indicating leaf health and photosynthetic capacity. SPAD measurements revealed variety A4 had the highest value (14.63). Values for A16 and A203 were significantly lower than A4, by 26.71% and 18.65%, respectively (Figure 2A). Chlorophyll content analysis corroborated this trend: A4 had the highest content, followed by A203, with A16 being the lowest. A16’s total chlorophyll content was 28.32% and 11.50% lower than A4 and A203, respectively (Figure 2B). The distribution patterns of chlorophyll a and b contents mirrored that of total chlorophyll across the varieties (Figure 2C,D). Variety A16 demonstrated the highest Pn, exceeding A4 and A203 by 22.40% and 27.87%, respectively (Figure 2E). Variety A203 exhibited the highest transpiration rate (Tr), surpassing A4 and A16 by 38.40% and 21.56% (Figure 2F). This higher transpiration rate facilitates water transport and cooling. Greater Cleaf promotes gas exchange, supplying more CO_2_ for photosynthesis. A16 had the highest Cleaf, significantly exceeding the other two varieties (Figure 2G). The intercellular CO_2_int was highest in A16, being 4.89% and 11.17% higher than in A4 and A203, respectively (Figure 2H). A203 showed the highest light use efficiency (LUE), significantly greater than the other varieties by 21.95% and 36.00% (Figure 2I). A16 had the highest water use efficiency (WUE), which was 32.73% and 78.81% higher than A4 and A203, respectively (Figure 2J). Leaf temperature did not differ significantly among varieties (Figure 2K). Stomatal impedance (Si) was lowest in A16, followed by A4 (not significantly different from A16), and was significantly highest in A203, exceeding A16 and A4 by 22.19% and 13.35%, respectively (Figure 2L).

### 3.3. Interspecific Differences in Leaf Stress Resistance Performance in Macadamia

Reactive Oxygen Species (ROS) are key signaling molecules in plant responses to stresses like drought, salinity, extreme temperatures, and pathogen infection. Their homeostasis directly impacts plant survival and adaptability [30,31]. DAB staining allows for the localization and semi-quantification of H_2_O_2_, where staining intensity and pattern indicate sites and relative abundance of H_2_O_2_ production; darker staining suggests potentially higher H_2_O_2_ levels. Assessment of stress resistance levels in seedlings of A4, A16, and A203 via DAB staining showed uniform coloration across leaves, with A4, A16, and A203 exhibiting similarly deep staining and little color difference (Figure 3A). Quantitative H_2_O_2_ analysis confirmed comparable H_2_O_2_ levels in seedlings of all three varieties (Figure 3B), consistent with DAB staining results. Conversely, O^2−^ quantification revealed the lowest O^2−^ accumulation in A203 leaves, while A16 seedlings had the highest levels, exceeding A4 and A203 by 64.39% and 358.89%, respectively (Figure 3C).

MDA, a product of membrane lipid peroxidation, indicates the degree of oxidative damage to cell membranes. SOD scavenges O^2−^ radicals. Higher CAT activity aids in decomposing hydrogen peroxide, mitigating oxidative damage [32]. MDA content analysis showed A16 had significantly higher levels than A4 and A203, by 53.92% and 132.35%, respectively, while A203 had the lowest MDA content (Figure 3D). Among the varieties, A4 exhibited the highest CAT activity, significantly greater than A16 and A203 by 137.14% and 139.82%, respectively (Figure 3E). A203 displayed the highest SOD activity, significantly exceeding A16 and A4 by 149.77% and 1333.01%, respectively (Figure 3F). POD activity in A203 seedling leaves was significantly lower than in A4 and A16, by 51.00% and 53.54%, respectively (Figure 3G).

### 3.4. Differences in Free Amino Acids in Leaves of Different Macadamia Varieties

High free amino acid content often indicates active nitrogen metabolism, reflecting the plant’s intrinsic metabolic level. Low content may suggest nitrogen deficiency or impaired metabolism. Analysis of free amino acids in seedling leaves of A4, A16, and A203 showed that A4 had significantly higher contents of Ala, Phe, and Pro than the other two varieties, exceeding A16 by 104.68%, 13.64%, and 42.87%, and A203 by 223.69%, 2165.22%, and 57.95%, respectively (Table 2). A16 seedling leaves showed significantly higher levels or were at relatively high levels for several amino acids including Asp, Ser, Glu, Val, Ile, Tyr, Lys, and Arg compared to the others. These were higher than A4 by 30.48%, 114.06%, 53.48%, 60.27%, 34.27%, 64.57%, 3.13%, and 12.44%, and higher than A203 by 22.22%, 161.11%, 20.91%, 97.70%, 17.36%, 92.36%, 24.31%, and 136.73%, respectively (Table 2). A203 seedling leaves had significantly higher Cys and Leu contents than the other varieties, exceeding A4 by 24.44% and 24.11%, and A16 by 1.64% and 11.92%, respectively. However, A203 had relatively low levels of Thr, Ala, His, and Arg among the three varieties (Table 2). Total free amino acid content was highest in A16 seedlings, intermediate in A4, and lowest in A203 (Table 2).

### 3.5. Correlation Between Leaf Phenotype and Physiological Indicators in Macadamia

A multi-dimensional correlation heatmap was constructed, encompassing morphological traits, photosynthetic physiology, the antioxidant system, and amino acid metabolism, to systematically elucidate interrelationships among these physiological processes. Correlation analysis revealed that photosynthetic physiological indicators, such as Pn and CO_2_int, showed highly significant positive correlations with various amino acids like Val, Ser, and Tyr. This reflects a synergistic mechanism where photosynthesis provides carbon skeletons and energy for amino acid synthesis, while amino acids participate in synthesizing key photosynthetic enzymes, thereby positively regulating photosynthetic efficiency (Figure 4). A cluster of positive correlations was observed among O^2−^, Val, Ser, Tyr, POD, Lys, and Arg. Positive correlations were also found between the osmoregulatory substances Pro and Ala with CAT, whereas weak negative correlations were noted for Lys and Met with some photosynthetic/antioxidant indicators (Figure 4). These results unveil cross-talk between osmoregulation and antioxidant defense, as well as resource allocation trade-offs for nitrogen among different metabolic pathways.

### 3.6. Interspecific Differences in Macadamia Fruit Phenotypes

Morphological analysis of fresh husked fruits from varieties A4, A16, and A203 was conducted. A16 fruits had a larger transverse diameter, while A4 fruits were smaller, being 13.79% and 16.99% smaller than A16 and A203, respectively (Figure 5A,B). A4 fruits had a significantly shorter longitudinal diameter compared to A16 and A203, by 17.90% and 19.55%, respectively (Figure 5A,C). A203 fruits were the heaviest, exceeding A4 and A16 by 58.86% and 4.33%, respectively. Both A16 and A203 had significantly higher fresh fruit weights than A4 (Figure 5D). Furthermore, A203 also had the heaviest fresh in-shell nut weight, surpassing A4 and A16 by 68.99% and 12.27%, respectively (Figure 5E,F). Analysis of the fruit husk showed that A16 and A203 had higher fresh husk weight and husk thickness, with no significant difference between these two varieties. A4 had the thinnest fresh husk, being 7.94% and 19.73% thinner than A16 and A203, respectively (Figure 5G,H). Differences in husk dry weight, husk water content, kernel recovery rate, and fruit shape index were also analyzed (Figure 5J–M). No significant difference was found in husk dry weight between A16 and A203, while A4’s husk dry weight was significantly lower than A16 and A203 by 56.88% and 51.24%, respectively (Figure 5J). A16 and A203 had the highest husk water content, whereas A4 had the lowest, containing 49.03% and 50.27% less than A16 and A203, respectively (Figure 5K). A203 showed a relatively higher kernel recovery rate, 7.38% and 8.17% higher than A4 and A16, although differences among varieties were not statistically significant (Figure 5L). The fruit shape indices were relatively similar, with A203 having the highest index, followed by A16, and A4 the lowest (Figure 5M).

### 3.7. Detection of Mineral Element Content in Macadamia Kernels

Analysis of mineral element content in kernels of A4, A16, and A203 revealed distinct varietal differences. For macronutrients, A4 kernels had the highest Ca content, exceeding A16 and A203 by 84.50% and 87.86%, respectively (Figure 6A). A203 kernels accumulated the highest levels of K and Mg, which were 110.98% and 102.57% higher than A4, and 91.49% and 94.98% higher than A16, respectively (Figure 6B,C). A16 kernels exhibited higher accumulation of Fe, Zn, and Cu, surpassing A4 by 21.44%, 15.17%, and 61.64% (Figure 6D–F), and A203 by 23.88%, 9.42%, and 1.59% (Figure 6D–F), respectively. Furthermore, A203 kernels had the highest Mn accumulation level, exceeding A4 and A16 by 82.23% and 66.09%, respectively (Figure 6G).

### 3.8. Comparison of Free Amino Acid Differences in Kernels of Different Macadamia Varieties

Free amino acid analysis in kernels of A4, A16, and A203 showed significant differences in the content of various free amino acids among the varieties (Table 3). A16 kernels had the highest total free amino acid content, which was 38.09% and 18.79% higher than A4 and A203, respectively. Specifically, A16 kernels contained higher levels of Thr, Ser, Gly, Tyr, Phe, His, Lys, Arg, and Pro, exceeding A4 by 61.59%, 8.92%, 64.33%, 176.66%, 125.75%, 288.17%, 165.15%, 623.92%, and 1661.40%, and surpassing A203 by 124.48%, 160.19%, 177.30%, 560.45%, 33.50%, 392.67%, 169.12%, 1006.57%, and 1712.27%, respectively (Table 3). A4 kernels had intermediate levels of many amino acids, with relatively high contents of Cys, Val, Met, Ile, and Leu. These were higher than A16 by 294.08%, 243.18%, 194.79%, 286.76%, and 347.20%, and higher than A203 by 62.21%, 59.70%, 299.38%, 136.84%, and 2.63%, respectively (Table 3). A203 kernels had higher contents of Asp, Glu, and Ala, which were 39.82%, 72.42%, and 73.77% higher than A4, and 34.32%, 135.84%, and 283.56% higher than A16, respectively (Table 3). Among the three varieties, Asp, Ser, and Glu were relatively abundant free amino acids, while Cys, Ile, and Leu were relatively scarce.

Subsequent classification analysis of free amino acids in the kernels revealed that A203 kernels had the highest content of medicinal amino acids (M), accounting for over 70% of the total amino acids (TAA). This was 17.56% and 25.40% higher than in A4 and A16 kernels, respectively (Appendix A). A4 kernels had the highest proportion of essential amino acids (E) within the TAA, exceeding A16 and A203 by 5.66% and 35.50%, respectively (Appendix A). A16 kernels contained more sweet, umami, bitter, and aromatic amino acids than the other two varieties. Specifically, sweet amino acids were 18.19% and 53.19% more abundant than in A4 and A203; umami amino acids were 2.57% and 58.13% more abundant; bitter amino acids were 15.86% and 132.19% more abundant; and aromatic amino acids were 74.98% and 57.19% more abundant, respectively (Appendix A).

### 3.9. Detection of Functional Substances in Macadamia Husks

The content of related phenolic substances was measured in the husks of varieties A4, A16, and A203. The content of p-hydroxybenzyl alcohol in A4 husks was significantly higher than in the other varieties, exceeding A16 and A203 by 63.86% and 14.76%, respectively (Figure 7A). Simultaneously, the content of 3,4-dihydroxybenzoic acid in A4 husks was significantly higher, surpassing A16 and A203 by 37.54% and 82.70%, respectively, with A203 having the lowest content (Figure 7B). The content of p-hydroxybenzoic acid in A4 husks was significantly higher, exceeding A16 and A203 by 83.56% and 135.05%, respectively, with A203 again having the lowest content (Figure 7C). The content of p-hydroxybenzaldehyde in A4 husks was significantly higher than in the other varieties, surpassing A16 and A203 by 197.82% and 107.79%, respectively, with A16 having the lowest content (Figure 7D).

### 3.10. Correlation Between Fruit Phenotype and Quality Indicators in Macadamia

Correlation analysis between fruit phenotypic and quality indicators for the three varieties was performed. Principal Component Analysis (PCA) indicated that the first principal component (PC1) explained 50.01% of the variance, and the second component (PC2) explained 39.95%. Together, they accounted for 89.96% of the total variance, indicating these two components efficiently and comprehensively captured the overall sample differences reflected by all measured indicators, representing the core information without needing additional components (Figure 8A). Regarding spatial distribution, A4 group samples were primarily located in the positive direction of both PC1 and PC2. A16 group samples clustered in the central area of the PCA plot, and A203 group samples tended towards the negative directions of PC1 and PC2. The three groups formed distinct and separate clusters with no significant overlap, visually demonstrating significant inter-group differences across multiple dimensions, including the compositional ratio of mineral elements, amino acid metabolic levels, fruit morphological characteristics, and the accumulation of phenolic substances.

Cluster analysis revealed a complex association network between external fruit traits and internal quality indicators, with numerous correlations reaching significant levels (denoted by asterisks in the figure). Firstly, among mineral elements, K and Mg showed significant positive correlations with husk thickness and nut longitudinal diameter, suggesting varieties rich in potassium and magnesium tend to develop fruits with thicker husks and greater longitudinal diameters. Mn exhibited strong positive correlations with multiple key phenotypic indicators, including husk thickness, fresh husked fruit transverse diameter, and fresh in-shell nut weight, highlighting its important role in promoting fruit enlargement and weight accumulation (Figure 8B). Secondly, among amino acid indicators, Lys and His showed significant positive correlations with phenotypic indicators such as fresh husk weight and husk water content, implying their accumulation might be closely related to water retention and biomass synthesis in the husk tissue. In contrast, indicators like Ca, phenol-3, Met, and Ile showed significant negative correlations with most phenotypic indicators. This reflects a potential resource allocation trade-off within the plant: when metabolic flux is directed towards accumulating certain mineral elements, phenolics, or specific amino acids, it might somewhat limit investment in the morphological growth of the fruit (Figure 8B).

Subsequent analysis of interrelationships among kernel quality components provided an in-depth examination of interactions between mineral elements, amino acids, and phenolic substances within the kernels. Several significant correlations revealed synergistic and antagonistic mechanisms. Among mineral elements, a strong positive correlation existed between K and Mg, indicating synergistic accumulation. In contrast, Fe showed a significant negative correlation with phenol-1, possibly suggesting chelation between iron ions and phenolic compounds or metabolic competition due to shared precursors. Regarding amino acid-phenolic associations, Cys showed a significant positive correlation with phenol-1, and Leu with phenol-4, providing clues that certain amino acids might serve as precursors for phenolic synthesis. Importantly, in cross-category associations, K showed significant positive correlations with Asp and Glu, the latter being umami amino acids. This explains, at a metabolic level, why high-potassium varieties might possess better flavor (as seen in A203). Concurrently, Fe showed widespread significant positive correlations with various amino acids including Lys, His, Arg, and Thr, strongly suggesting iron, possibly as a cofactor or regulatory factor, is widely involved in nitrogen metabolism and amino acid synthesis pathways within the kernels. These significant correlation networks provide key theoretical targets for synergistically regulating mineral nutrition to improve kernel nutritional and flavor quality (Figure 8C).

## 4. Discussion

It is important to note that this study was conducted under the specific pedo-climatic conditions of the experimental site in Pu’er, Yunnan. While this provides a unified context for comparing the relative performance of the three varieties, the absence of detailed soil profiles and continuous microclimatic data limits the extrapolation of the absolute values of these physiological and quality traits to other environments. Future studies incorporating multi-environment trials will be valuable to dissect the genotype-by-environment interactions. Nonetheless, the observed differences robustly reflect the distinct genetic backgrounds and physiological strategies of the varieties under the same managed conditions.

### 4.1. Analysis of Apparent Differences Among Macadamia Varieties

When plants face transplanting stress, their internal homeostasis is altered [33], leading to growth arrest, abnormal growth states, or even death, significantly hindering survival and recovery post-transplantation [34]. During transplanting, plant root systems inevitably suffer some damage, causing changes in multiple physiological indicators of seedlings. Therefore, monitoring these physiological changes can effectively track plant growth status. Leaf phenotypic traits reflect plant growth and development, with significant characteristic differences existing among varieties [35]. Leaves are crucial organs for carbon assimilation, and their morphology affects the photosynthetic area and LUE. This, in turn, determines various plant functions, including photosynthesis, transpiration, nutrient requirements, and adaptation to extreme environments. Leaves influence growth through physiological, biochemical, morphological, or developmental mechanisms, embodying plant survival strategies under natural selection [36].

The analysis of apparent leaf differences among macadamia varieties revealed rich diversity in their morphology, photosynthesis, and physiological characteristics. The results of this study indicate significant distinctions in leaf morphology among the tested varieties. Variety A4 had yellowish-green young leaves maturing to green, ovate blades, entire margins with numerous marginal spines, an acute apex, an acuminate base, and ternate whorled phyllotaxy. Variety A16 had slightly red young leaves turning light green upon maturation, obovate blades, undulate margins with few spines, an obtuse apex, an acute base, and both ternate and quaternate whorled characteristics. Variety A203 had green young leaves becoming dark green at maturity, oblanceolate blades, prominently undulate margins with few spines, a sharp-acuminate apex, an acuminate base, and also exhibited ternate and quaternate whorled phyllotaxy (Table 1, Figure 1). The studied leaf morphologies showed clear differences, which are crucial for plant classification and identification [37]. Research by Sun et al. also confirmed that leaf morphology is an important indicator for plant classification [38], underscoring its significance in variety classification.

The results also showed significant differences in photosynthetic efficiency and related physiological indicators among the varieties. Variety A4 had higher SPAD values and chlorophyll content, while variety A16 exhibited the highest Pn, Cleaf, and WUE. Variety A203 demonstrated higher Tr and LUE (Figure 2). Studies indicate that chlorophyll a content, chlorophyll b content, PSII potential activity (Fv/Fo), and the primary photochemical efficiency of PSII (Fv/Fm) show highly significant or significant differences among different macadamia varieties. Such interspecific differences in photosynthetic characteristics reflect divergent strategies in light capture and utilization among varieties.

### 4.2. Analysis of Physiological Differences Among Macadamia Varieties

The seedling selection in this study was based on an integrated model considering morphology, photosynthesis, and oxidative balance. The leaf morphological and structural characteristics of different varieties determine their survival capacity under specific environmental conditions. Traits such as leaf margin morphology, spine number, leaf base shape, and leaf shape index further influence the seedlings’ adaptability to environmental changes. Therefore, selecting plant varieties suitable for specific ecological niches should fully consider these morphological characteristics and their impact on seedling adaptability. High photosynthetic rate, strong antioxidant enzyme (SOD, CAT, POD) activity, low levels of harmful substances (MDA, H_2_O_2_, O^2−^), and appropriate free amino acid levels are beneficial for plant adaptability. Under new environmental conditions, maintaining good photosynthetic performance and stable enzyme activity indicates stronger post-transplant adaptability for a given variety. These three varieties exhibited different regulatory modes for adapting to the new environment. Differences in H_2_O_2_ and O^2−^ content among the varieties were observed, with DAB staining results consistent with H_2_O_2_ content measurements, reflecting differing physiological traits related to oxidative stress among varieties. The distinct configurations of the antioxidant enzyme system (CAT, SOD, POD) provide a mechanistic basis for interpreting the observed ROS signaling and associated metabolic regulation across the varieties. In A4, the significantly highest CAT activity, coupled with the lowest MDA content, indicates a highly efficient mechanism for H_2_O_2_ detoxification. This robust CAT-mediated clearance likely maintains H_2_O_2_ below damaging thresholds, yet potentially permits its function as a signaling molecule within a sub-toxic range, fostering a cellular environment conducive to the observed accumulation of osmoprotectants like Pro and Ala. This enzymatic strategy underpins A4’s constitutive stress tolerance. Conversely, A203’s profile was dominated by the highest SOD activity, facilitating the rapid dismutation of superoxide (O^2−^), which aligns with its lowest O^2−^ content. This prioritization of primary ROS conversion at the source minimizes the generation of more reactive secondary species and reduces the metabolic load on downstream scavengers like POD (which was lowest in A203). This efficient O^2−^ management likely contributes to membrane stability (low MDA) and supports high light-use efficiency by mitigating photo-oxidative damage. The coordinated yet distinct activities of these antioxidant enzymes in each variety not only define their capacity to scavenge specific ROS but also shape the spatiotemporal dynamics of the ROS landscape (e.g., H_2_O_2_/O^2−^ ratio), which are known to differentially modulate transcription factors and calcium signaling, thereby influencing the expression of metabolic genes. This could explain the varietal divergence in nitrogen assimilation and amino acid partitioning, such as the high nitrogen flux towards Asp, Ser, and Glu in A16 versus the preferential accumulation of specific osmolytes in A4. Thus, the genotype-specific blueprint of the antioxidant enzyme network is a fundamental regulator that connects ROS metabolism to broader metabolic reprogramming and stress adaptation strategies.

Higher POD activity is also beneficial for coping with stress. Comparing CAT and SOD activities suggests that A4 has an advantage in dealing with oxidative stress, resulting in relatively less cellular damage post-transplantation. A203 demonstrated a relatively strong ability to resist oxidative stress after transplantation. Analysis of MDA content led to the conclusion that A203 has relatively better cell membrane stability compared to the other two varieties, while A16 performed moderately. Considering photosynthetic indicators comprehensively, variety A16 excelled in net photosynthetic rate, intercellular CO_2_int, and WUE; A203 performed excellently in LUE; variety A4’s leaf photosynthetic capacity under the new environment might be at a similar level.

Following the transplantation of macadamia seedlings, during their habitat adaptation process, indicators such as leaf phenotypic traits, photosynthetic rate, antioxidant enzymes, and the contents of MDA, H_2_O_2_, O^2−^, and free amino acids can, to some extent, quantitatively reflect the degree of seedling adaptation to the new environment. Leaf phenotypic traits influence photosynthetic rate through factors like chlorophyll content and adapt to adverse weather by altering leaf shape index and marginal spine number. Photosynthetic rate directly affects energy acquisition and substance synthesis, providing the material and energy foundation for seedling growth and development in the new environment. Antioxidant enzymes (e.g., SOD, POD, CAT) constitute the plant’s antioxidant defense system. When seedlings face various stresses due to transplanting, these enzymes promptly scavenge excess ROS, maintaining intracellular redox balance, mitigating oxidative damage, and assisting seedling adaptation. MDA, H_2_O_2_, and O^2−^, as oxidative products, reflect the degree of oxidative damage in seedling cells. Fluctuations in their content within a certain range represent normal plant responses to environmental changes, but excessively high levels indicate severe oxidative stress, potentially affecting normal growth and survival. Free amino acids play a key role in metabolism and adaptation to the new environment. Metabolically, they can serve as respiratory substrates for energy supply and participate in protein synthesis and repair of damaged tissues. In terms of adaptability, they help regulate osmotic pressure in response to environmental water changes and can act as signaling molecules involved in stress responses, enhancing resistance to biotic and abiotic stresses, thereby improving post-transplant survival rates and growth conditions.

Correlation analysis between leaf phenotype and physiological traits indicated that SOD is associated with changes in oxidative products when coping with oxidative stress, reflecting its role in regulating their levels. POD functions in scavenging MDA and other oxidative products, showing a close link between them in the antioxidant defense process. CAT not only participates in antioxidation but may also play a role in regulating amino acid metabolism, indicating a potential connection with amino acid metabolism. The analysis also revealed close linkages between photosynthetic rate, gas exchange, and LUE, collectively influencing photosynthetic efficiency. Transpiration and leaf temperature regulation interact mutually, functioning synergistically to maintain plant water balance and temperature stability. Most free amino acids showed varying degrees of correlation with each other, reflecting their interconnections in nitrogen metabolism and substance synthesis, likely participating collectively in the regulation of plant growth and development. Differences in the correlations among indicators across varieties suggest that varietal characteristics may influence the interrelationships of these physiological parameters, providing clues for studying differences in physiological traits among varieties. These correlations help understand the intrinsic links among various physiological processes in macadamia seedlings, offering theoretical references for variety selection and cultivation management optimization.

The distinct biochemical profiles observed among the varieties have direct ecological consequences for their adaptability and potential niche differentiation. The superior antioxidant capacity (high CAT, low MDA) in A4 signifies a robust constitutive defense system, an advantageous strategy for pre-adaptation to environments prone to frequent abiotic stresses such as high light, temperature fluctuations, or poor soil conditions. This biochemical investment in stress mitigation likely underlies its reputation for strong cold and wind tolerance. In contrast, A203’s strategy of high light use efficiency (LUE) coupled with efficient O^2−^ scavenging (high SOD, low O^2−^ content) suggests an adaptation optimized for maximizing carbon gain under moderate stress levels, potentially favoring competitive growth in stable but light-limited environments. Furthermore, the high nitrogen metabolic activity (reflected by high total free amino acid content) in A16 aligns with a resource-acquisitive strategy, supporting rapid growth and high yield under favorable conditions with ample nutrient and water availability. Thus, the interspecific variation in key biochemical traits—antioxidant systems, photosynthetic efficiency, and nitrogen metabolism—not only explains the differential physiological performance but also predicts their ecological success and suitable habitats, providing a mechanistic basis for strategic cultivar deployment in diverse agro-ecological zones.

### 4.3. Correlation Analysis Between Leaf Phenotype and Physiological Status in Macadamia

The leaf shape index can significantly influence photosynthetic capacity. Research on cassava (*Manihot esculenta* Crantz) found that changes in leaf morphology lead to alterations in photosynthetic parameters, indicating the important role of leaf morphology in enhancing photosynthetic efficiency [39]. This finding is consistent with observations in poplar under different soil water conditions, which suggested that a larger leaf shape index contributes to improved photosynthesis and WUE [40]. Furthermore, the absolute dimensions of leaf width and length also play crucial roles in light capture and transpiration regulation. Studies have shown that increased leaf area can raise the upper limit of the photosynthetic rate; however, photosynthetic efficiency declines when excessive self-shading occurs due to overly large leaf area. Additionally, the leaf shape index indirectly affects the antioxidant system. Leaf morphology determines the distribution of photosynthetic light intensity; excessively high light intensity readily generates excess ROS, thereby inducing increased activities of SOD, CAT, and POD. Research indicates that crops with more elongated leaves exhibit higher SOD and CAT activities under high light stress to mitigate ROS-induced membrane lipid peroxidation (MDA) accumulation [41]. The leaf shape index serves as a key bridge connecting leaf morphology with photosynthetic, transpirational, and antioxidant functions. In this study, leaves of the three different macadamia varieties exhibited clear differences in characteristics such as shape and margin, whereas differences in leaf length, width, and shape index were relatively less pronounced. Concurrently, significant differences existed in physiological indicators such as photosynthetic efficiency and antioxidant indices among the varieties. Research on cotton (Gossypium spp.) demonstrated that when leaf morphological differences are not significant, significant differences in MDA and antioxidant enzymes like SOD can serve as key indicators for assessing variety adaptability [42]. Similarly, studies on lettuce (*Lactuca sativa* L.) indicated that when morphology shows no significant change, MDA and SOD levels can directly reflect the plant’s stress level and tolerance [43]. These studies demonstrate that the strength of the antioxidant system directly determines the level of MDA, thereby influencing the duration of effective photosynthesis and yield stability. In variety selection and stress assessment, simultaneously measuring indicators such as leaf morphology, MDA, SOD (and other antioxidant enzymes) can provide a more comprehensive evaluation of adaptability.

### 4.4. Analysis of Interspecific Differences in Macadamia Fruit Appearance

Studies on the apparent differences among nut species often utilize basic morphological indicators such as the diameter, longitudinal diameter, and weight of the fresh husked fruit. The kernel recovery rate is further calculated based on the weight of the shelled nut and the fresh husked fruit weight. Husk thickness, fresh husk weight, and dry husk weight can be used to estimate husk water content, thereby assessing water storage characteristics across different species. The fruit shape index is an important parameter describing fruit geometry, commonly used to distinguish morphological categories like oblate, spherical, ovoid, and olive-shaped. Research on different provenances of oil-tea camellia (*Camellia oleifera*) indicated that a fruit shape index between 0.89 and 1.07 corresponds to a spherical shape, and the kernel recovery rate varied significantly among provenances, revealing notable differences in fruit structure and water allocation [44]. Studies on different hazelnut (*Corylus* spp.) varieties found a large coefficient of variation for kernel recovery rate, indicating that the interspecific genetic background significantly influences the proportion of the edible portion [45].

Internal structural characteristics like kernel recovery rate more directly impact the practical utilization value of crop fruits, especially in nuts and seed crops. Traits with high coefficients of variation provide greater selection potential for breeding and can serve as stable traits for variety identification. In this study, distinct differences in fruit morphology were observed among the different macadamia varieties, particularly between A4 and the other two varieties (Figure 5). This provides a reliable reference for variety evaluation and for determining directions for production and processing development. Fruit size assessment is necessary for establishing quality standards, enhancing market value, monitoring fruit growth, predicting yield, and determining optimal fertilization and irrigation levels. It also holds significant importance for the design and development of size-sorting equipment [46]. Within the macadamia industry, morphological indicators of fruit size are crucial factors determining fruit value and processing methods; larger fruits generally command higher economic value. Conversely, varieties with higher husk yield can gain advantages in husk processing applications. Macadamia fruit size, shape, and shell thickness directly influence commercial grading and economic value. Macadamia morphological data can also serve as an objective basis for grading standards, providing a theoretical foundation for variety selection and production processing methods for different macadamia varieties.

### 4.5. Analysis of Interspecific Differences in Macadamia Fruit Quality

Mineral elements are important components of the nutritional value of nuts. Studies have shown that eight elements—K, Mg, P, Ca, Mn, Fe, Zn, Cu—exhibit significant differences among different macadamia varieties, with generally large coefficients of variation. Potassium (K) content is the highest, and manganese (Mn) shows the greatest variation, indicating its significant contribution to nutritional differences among varieties [7]. In this study, Ca, K, Mg, Fe, Zn, Cu, and Mn exhibited distinct accumulation patterns in the kernels of the three varieties A4, A16, and A203 (Figure 6). These differences are closely related to soil nutrient supply and fertilization management, particularly as total nitrogen and phosphorus levels directly promote the accumulation of lipids and proteins. Therefore, when breeding varieties with high mineral nutritional value, indicators such as K, Mn, and Ca can be incorporated into multi-trait selection indices to achieve concurrent improvement in both nutrition and yield.

The amino acid composition determines the nutritional value of nuts. Amino acid analysis of various edible nuts revealed that essential amino acids account for approximately 27–28% of the TAA in macadamia [47]. In this study, variety A4 had the highest proportion of essential amino acids compared to A16 and A203 (Table 3). Among non-essential amino acids, hydrophobic amino acids showed the highest proportion (approx. 37–44%), followed by acidic amino acids (approx. 28–33%). While the proportion of essential amino acids differed little among varieties, significant differences existed in the content of non-essential amino acids, directly affecting the protein’s biological value and flavor characteristics [47]. In this study, variety A16 kernels had a high total free amino acid content, with umami amino acids constituting 36%, indicating pronounced flavor characteristics (Table 3). These findings provide a molecular basis for varietal stratification using amino acid profiles. The amino acid composition of macadamia is predominantly medicinal amino acids. In variety A203, medicinal amino acids accounted for over 70% of the TAA (Appendix A), while the essential amino acid content was relatively the lowest.

The distinct amino acid profiles of the kernels reveal clear directions for targeted industrial applications. The high abundance of umami and sweet amino acids (e.g., Glu, Asp, Ala, Gly, Ser) in A16 kernels directly contributes to a superior, inherent savory and sweet taste profile. This makes A16 particularly suitable for high-value markets such as premium fresh consumption, gourmet snacks, and as a natural flavor enhancer in the culinary and food manufacturing industries, potentially reducing the need for added monosodium glutamate or other synthetic flavorings. In contrast, A203, with its exceptionally high proportion of medicinal amino acids (e.g., specific profiles of Arg, Leu, Ile), holds significant promise for the nutraceutical and functional food sectors. Kernels or extracts from A203 could be developed into targeted nutritional supplements aimed at supporting specific physiological functions, such as immune modulation or metabolic health. Furthermore, the unique and balanced amino acid composition across all varieties underscores their high value as a source of plant-based proteins for the health food market.

Beyond nutritional composition, pollination method represents a fundamental biological factor influencing fruit development and quality. As documented in the plant materials, the three varieties exhibit distinct pollination characteristics: A4 is highly self-sterile, while both A16 and A203 show a tendency for cross-pollination. These differential pollination behaviors may contribute to the observed variations in fruit set efficiency, kernel development, and ultimately, the accumulation of specific quality components [14]. In macadamia, cross-pollination has been associated with improved kernel recovery rate and more uniform nut development, which aligns with the superior fruit weight and morphological consistency observed in A16 and A203 compared to the highly self-sterile A4. Future studies specifically designed to dissect the pollination regime’s effect on metabolic profiles (e.g., amino acids, lipids) of these varieties would provide deeper insights into the interaction between reproductive biology and nut quality formation.

Furthermore, as noted in the plant materials, varietal differences in pollination type (e.g., the high self-sterility of A4) may have also contributed to the observed variations in fruit set and kernel quality parameters. While the experimental orchard provided opportunities for cross-pollination, the inherent pollination efficiency and compatibility of each variety represent an intrinsic trait that influences yield and quality formation, and should be considered in future breeding and orchard management strategies.

The husk is the primary waste product generated during macadamia processing, but it is rich in active components such as polyphenols, tannins, and soluble sugars, possessing potential for developing high-value-added products. Research indicates that 50% ethanol most effectively extracts phenolic and flavonoid compounds, and encapsulating these substances using liposomes can form a stable and protective system [48]. The high soluble sugar content in husks allows for the production of nitrogen-, phosphorus-, and potassium-rich organic fertilizer through fermentation, enhancing soil fertility and enabling resource recycling [49]. Furthermore, shells subjected to magnetic nano-processing can serve as magnetic adsorbents for organic pollutant recovery, or yield silicon-carbon composite materials after high-temperature carbonization for use in high-performance ceramics or electrodes, demonstrating potential for a full-chain transformation from waste to functional material. Additionally, macadamia husks and shells have direct application value in antioxidants, natural pesticides, and organic fertilizer production.

In this study, the accumulation of phenolic substances like 3,4-dihydroxybenzoic acid and p-hydroxybenzoic acid in the husks of variety A4 was significantly higher than in A16 and A203, indicating high antioxidant potential (Figure 7). The distinct amino acid profiles of the kernels reveal clear directions for targeted industrial applications. The high abundance of umami and sweet amino acids (e.g., Glu, Asp, Ala, Gly, Ser) in A16 kernels directly contributes to a superior, inherent savory and sweet taste profile. This makes A16 particularly suitable for high-value markets such as premium fresh consumption, gourmet snacks, and as a natural flavor enhancer in the culinary and food manufacturing industries, potentially reducing the need for added monosodium glutamate or other synthetic flavorings. In contrast, A203, with its exceptionally high proportion of medicinal amino acids (e.g., specific profiles of Arg, Leu, Ile), holds significant promise for the nutraceutical and functional food sectors. Kernels or extracts from A203 could be developed into targeted nutritional supplements aimed at supporting specific physiological functions, such as immune modulation or metabolic health. Furthermore, the unique and balanced amino acid composition across all varieties underscores their high value as a source of plant-based proteins for the health food market.

### 4.6. Correlation Analysis Between Macadamia Fruit Appearance and Quality

In this study, amino acid indicators such as Lys and His showed significant positive correlations with several phenotypic traits, including fresh husk weight and husk water content. In contrast, Ca, phenol-3, Met, Ile, phenol-4, phenol-2, Cys, and Val showed significant negative correlations with most phenotypic indicators (Figure 8B). These results indicate complex correlations between macadamia fruit traits and nutritional components, with rich variation and multidirectionality in fruit quality traits. Free amino acids act as flavor precursors in food; their participation in the Maillard reaction with reducing sugars can form aromatic components like pyrazine heterocycles, furan derivatives, and pyrroles. Concurrently, specific amino acids such as phenylalanine undergo decarboxylation and redox reactions during thermal processing, generating volatile compounds like phenethyl alcohol (rosy aroma) and benzyl alcohol (almond-like aroma). These reaction products synergistically influence sensory quality, playing a significant regulatory role in the formation of food flavor characteristics [50]. In this study, variety A16 kernels contained higher levels of sweet, umami, bitter, and aromatic amino acids compared to the other two varieties, suggesting potentially richer flavor profiles (Appendix A). Hydrolyzed amino acids [51] and mineral elements [52] are also essential for human nutrition. Variety A203 had the highest levels of mineral elements (K, Mg, Mn) and the highest total hydrolyzed amino acids (Figure 6, Table 3). The levels of these two indicators are key factors determining final fruit quality. Ultimately, the synergistic consideration of kernel amino acids and husk phenolics enables a whole-industry-chain optimization strategy. For instance, A16 fruits, with their superior kernel flavor, should be directed towards the fresh nut and high-end confectionery markets to maximize economic return. Simultaneously, A4, with its smaller fruit size but phenolics-rich husk, presents a compelling case for integrated processing where the primary product (kernel) and the by-product (husk) are both efficiently utilized. This dual-value approach not only enhances the economic resilience of macadamia cultivation but also aligns with circular bio-economy principles. Our study provides the scientific rationale for such varietal specialization in breeding programs and industrial processing.

Fruit quality selection involves constructing a trait-quality association model to establish a dual-cycle evaluation system encompassing nutritional function and processing characteristics. The relationship between plant fruit morphology and edible traits is very close [53]. Morphological indicators contribute significantly to fruit quality assessment and largely determine the commercial value of macadamia. Nutritional components and taste are also crucial factors considered during variety selection [53,54]. This study analyzed the nutritional components of kernels from different macadamia varieties and incorporated husk phenolics into the variety selection system, thereby expanding the pathways for valorizing macadamia by-products with unique quality traits. Macadamia primarily relies on its fruit as the economic product and distinctive advantage. Fruit size, weight, kernel recovery rate, etc., directly determine its commercial value. The content of nutritional components in the kernel is closely linked to consumer demand. Analyzing morphological indicators and nutritional components directly reflects the quality of macadamia fruits. Conducting such measurements and analyses for fruits from different varieties enables a comprehensive assessment of quality differences, guiding commercial classification and processing, facilitating targeted utilization of fruits from specific varieties, and enhancing the economic value of the nuts. Furthermore, significant correlations exist between the apparent traits of macadamia fruits (e.g., size, weight, kernel recovery rate) and their intrinsic nutritional quality. Morphological indicators not only directly determine commercial value but also, together with kernel nutritional components, form the core basis for quality evaluation. By constructing trait-quality association models, varieties possessing both excellent edible traits and nutritional functions can be systematically selected. Simultaneously, incorporating by-products like husk phenolics into the evaluation system expands the avenues for high-value utilization. This provides scientific theoretical support and practical guidance for targeted breeding, commercial classification, processing, and enhancing the economic benefits of the industry.

## 5. Conclusions

This study systematically compared the differences in leaf physiological characteristics at the seedling stage and fruit quality at the mature stage among macadamia varieties A4, A16, and A203, revealing varietal specificities in morphology, photosynthesis, antioxidant capacity, amino acid metabolism, and mineral element accumulation. The results demonstrate that each variety possesses a unique combination of traits, forming specific adaptation and quality profiles: A4 exhibited outstanding performance in chlorophyll content, antioxidant enzyme activity, and husk phenolic compound accumulation, which underscores its strong stress resistance potential and high value for by-product utilization. A16 showed significant advantages in photosynthetic efficiency, water use efficiency, and total amino acid content in kernels, particularly its richness in flavor amino acids, making it highly suitable for fresh consumption and high-value product development. A203 excelled in light use efficiency, accumulation of mineral elements (such as K, Mg, Mn), and the proportion of medicinal amino acids, demonstrating superior potential for nutritional functionality.

Through multi-dimensional correlation analysis and principal component analysis, this study further elucidated the intrinsic relationships between leaf phenotype and physiological status, as well as the complex regulatory network between fruit morphology and intrinsic quality. More importantly, we established a comprehensive multi-trait evaluation system encompassing “morphology—photosynthesis—antioxidant activity—amino acids—quality,” which provides a scientific framework for variety assessment. The identified variety-specific trait profiles provide clear targets for future breeding programs, enabling the selection of parental lines for developing new cultivars with desired stress resistance, flavor, or nutritional traits. Furthermore, the physiological adaptation strategies revealed at the seedling stage offer valuable criteria for designing regional adaptation experiments to match specific varieties with optimal growing environments.

In practical applications, these findings offer crucial guidance for targeted variety selection and cultivation management: A4 is recommended for cultivation under adverse conditions or for by-product utilization; A16 is ideal for fresh consumption and high-nutritional-value product development; while A203 shows great promise for nutritionally fortified products. The comprehensive evaluation system established in this study not only provides a theoretical basis for macadamia variety breeding and whole-industry-chain development but also lays a scientific foundation for promoting sustainable development and value-added utilization of the nut industry.

## Figures and Tables

**Figure 1 biology-14-01638-f001:**
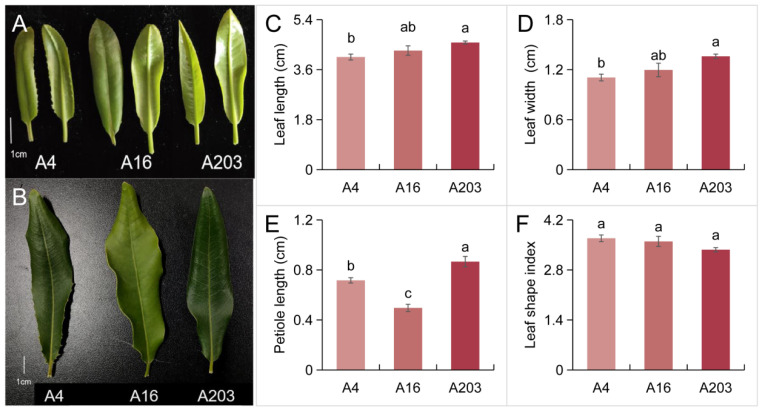
Analysis of phenotypic differences in *Macadamia integrifolia* leaves. (**A**) Young leaves of cultivars A4, A16, and A203 (bar = 1 cm). (**B**) Mature leaves of cultivars A4, A16, and A203 (bar = 1 cm). (**C**) Leaf length. (**D**) Leaf width. (**E**) Petiole length. (**F**) Leaf shape index. The results shown are the means ± SDs (*n* = 3), and different letters indicate significant differences (*p* < 0.05 according to Tukey’s test).

**Figure 2 biology-14-01638-f002:**
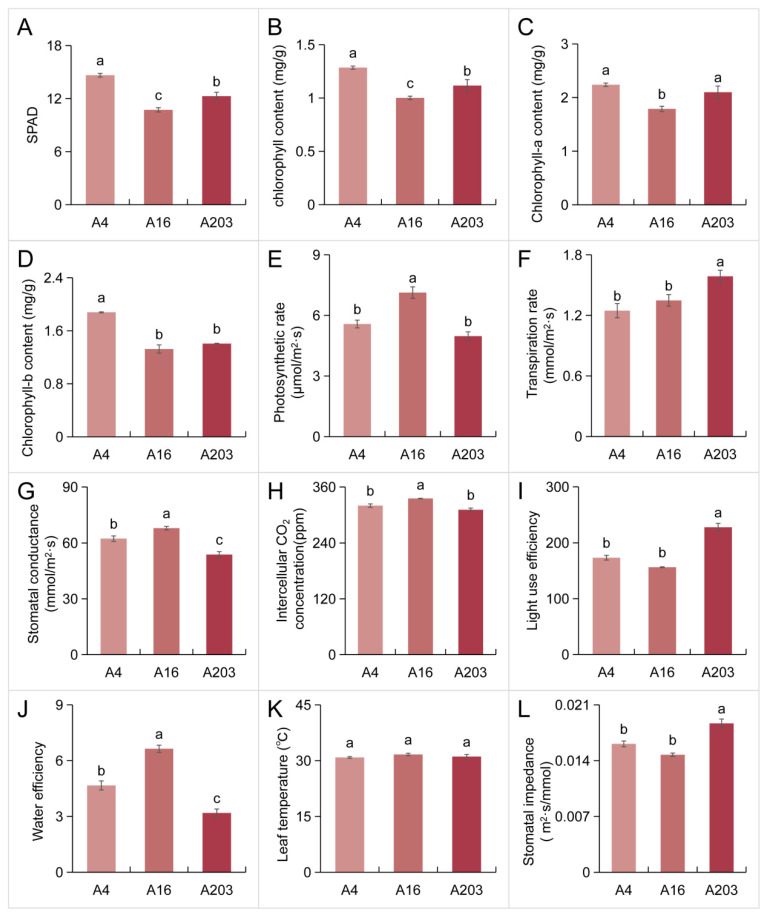
Analysis of photosynthetic efficiency parameters of *Macadamia integrifolia* leaves. (**A**) SPAD value. (**B**) Total chlorophyll content. (**C**) Chlorophyll-a content. (**D**) Chlorophyll-b content. (**E**) Net Photosynthetic rate (Pn). (**F**) Transpiration rate (Tr). (**G**) Stomatal conductance (Cleaf). (**H**) Intercellular CO_2_ concentration (Ci). (**I**) Light use efficiency (LUE). (**J**) Water efficiency (WUE). (**K**) Leaf temperature (Tleaf). (**L**) Stomatal impedance (Si). The results are presented as the means ± SDs (*n* = 3). Different letters indicate significant differences (*p* < 0.05 according to Tukey’s test).

**Figure 3 biology-14-01638-f003:**
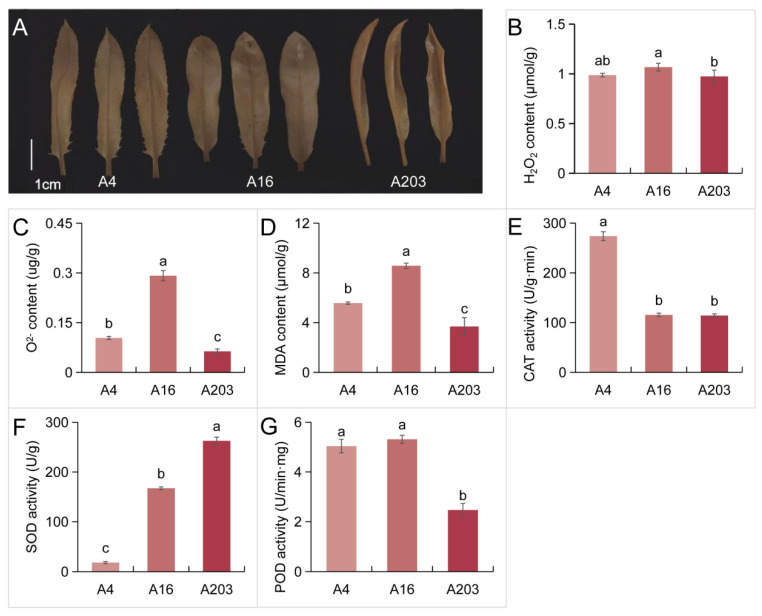
Comparative analysis of the stress resistance performance of *Macadamia integrifolia* leaves. (**A**) Comparison of DAB staining in young leaves of cultivars A4, A16, and A203, (**B**) H_2_O_2_ content, (**C**) O^2−^ content, (**D**) MDA content, (**E**) CAT activity, (**F**) SOD activity, (**G**) POD activity. The results are presented as the means ± SDs (*n* = 3). Different letters indicate significant differences (*p* < 0.05 according to Tukey’s test).

**Figure 4 biology-14-01638-f004:**
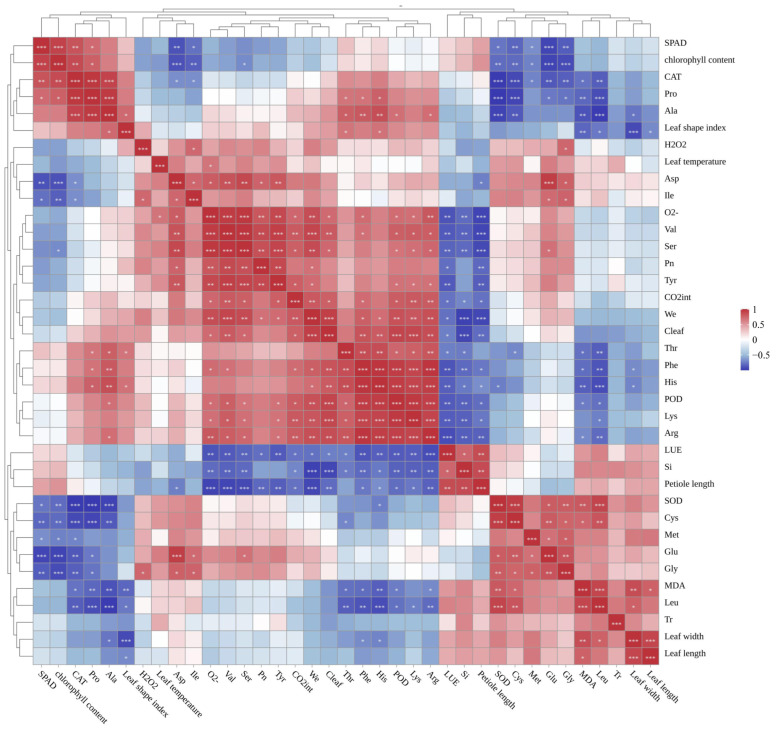
Correlation analysis of leaf phenotypes and physiological indicators of *Macadamia integrifolia*. Dark blue represents strong negative correlation, white represents no correlation, and dark red represents strong positive correlation. *** indicates extremely significant correlation (*p* < 0.001), ** indicates highly significant correlation (0.001 ≤ *p* < 0.01), and * indicates significant correlation (0.01 ≤ *p* < 0.05).

**Figure 5 biology-14-01638-f005:**
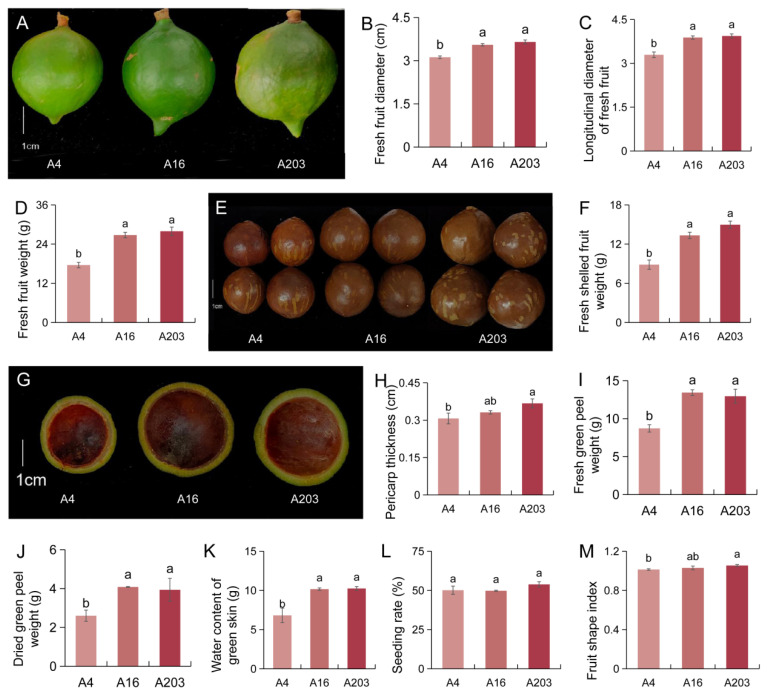
Analysis of phenotypic differences in *Macadamia* nuts. (**A**) Comparison of fresh fruits of cultivars A4, A16, and A203. (**B**) Fresh fruit diameter (horizontal diameter). (**C**) Fresh fruit longitudinal diameter. (**D**) Fresh fruit weight. (**E**) Comparison of fresh pericarps of cultivars A4, A16, and A203. (**F**) Fresh shelled fruit weight. (**G**) Fresh husk of cultivars A4, A16, and A203. (**H**) Pericarp thickness. (**I**) Fresh green peel weight. (**J**) Dried green peel weight. (**K**) Water content of green skin. (**L**) Seeding rate. (**M**) Fruit shape index. The results are presented as the means ± SDs (*n* = 3). Different letters indicate significant differences (*p* < 0.05 according to Tukey’s test).

**Figure 6 biology-14-01638-f006:**
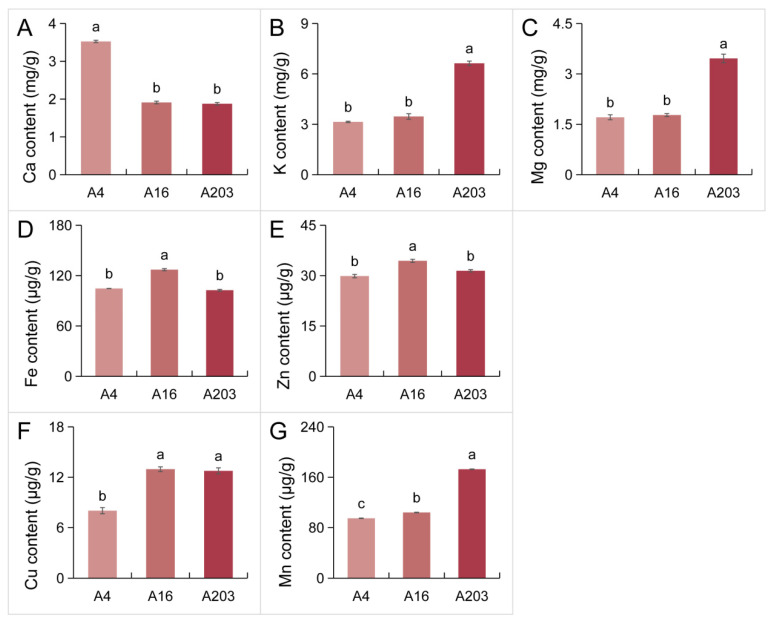
Detection and analysis of mineral element content in *Macadamia* nut kernels. (**A**) Ca content, (**B)** K content, (**C**) Mg content, (**D**) Fe content, (**E**) Zn content, (**F**) Cu content, (**G**) Mn content. The results are presented as the means ± SDs (*n* = 3). Different letters indicate significant differences (*p* < 0.05 according to Tukey’s test).

**Figure 7 biology-14-01638-f007:**
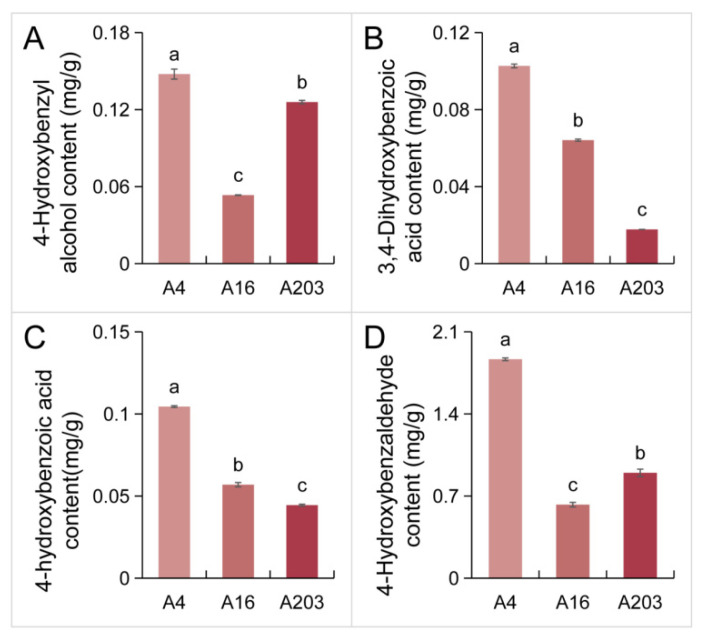
Detection and analysis of functional substances in *Macadamia* nut peels. (**A**) 4-Hydroxybenzyl alcohol content, (**B**) 3,4-Dihydroxybenzoic acid content, (**C**) 4-hydroxybenzoic acid content, (**D**) 4-Hydroxybenzaldehyde content. The results are presented as the means ± SDs (*n* = 3). Different letters indicate significant differences (*p* < 0.05 according to Tukey’s test).

**Figure 8 biology-14-01638-f008:**
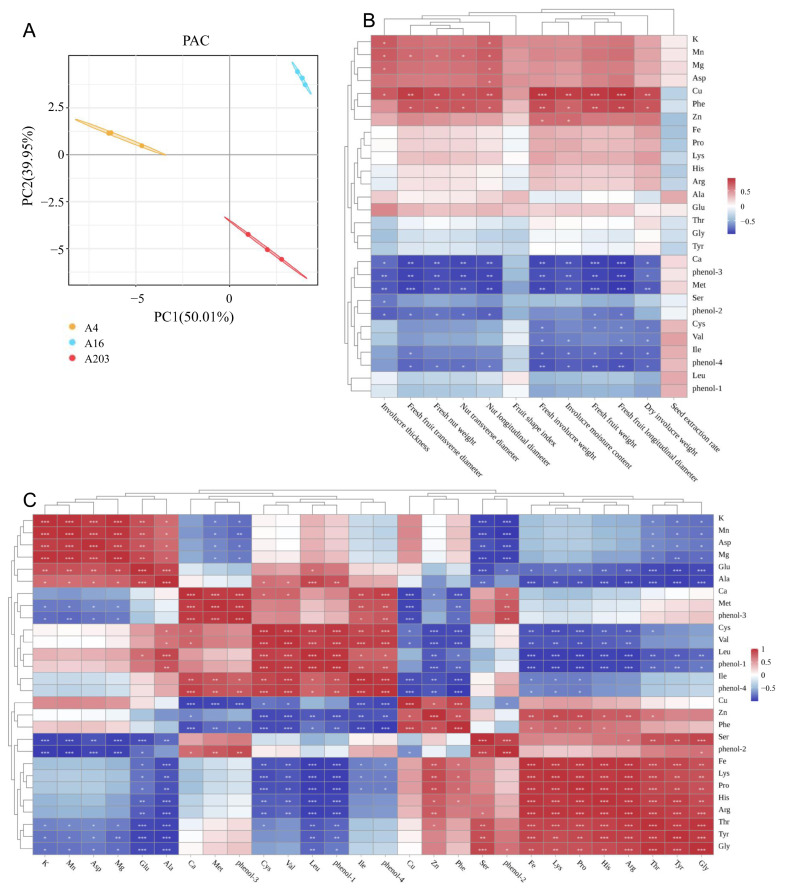
Correlation analysis of fruit phenotypes and quality indicators of *Macadamia* nuts. (**A**) Principal Component Analysis (PCA) score plot showing the separation of the three varieties (A4, A16, A203) based on combined fruit phenotype and quality data. (**B**) Heatmap of inter-group correlations between fruit phenotypic indicators (columns) and kernel quality indicators (rows). (**C**) Heatmap of intra-group correlations among kernel quality indicators (mineral elements, amino acids, phenolic compounds). In panels (**B**,**C**), the phenolic compounds are labeled as Phenol-1 to Phenol-4, corresponding to p-hydroxybenzyl alcohol, 3,4-dihydroxybenzoic acid, p-hydroxybenzoic acid, and p-hydroxybenzaldehyde, respectively. Dark blue represents strong negative correlation, white represents no correlation, and dark red represents strong positive correlation. *** indicates extremely significant correlation (*p* < 0.001), ** indicates highly significant correlation (0.001 ≤ *p* < 0.01), and * indicates significant correlation (0.01 ≤ *p* < 0.05).

**Table 1 biology-14-01638-t001:** Comparison of the apparent morphology of *Macadamia integrifolia* leaves.

Traits	A4	A16	A203
Phyllotaxy	3-whorled	3–4 whorled	3–4 whorled
Leaf shape	Ovate	Obovate	Oblanceolate
Leaf apex shape	Acute	Obtuse	Sharply acute
Leaf base shape	Acuminate	Acute	Acuminate
Leaf margin shape	Smooth	Undulate	Distinctly undulate
Spines on leaf margin	more	few	few
Young leaf color	Yellowish-green	Reddish	Green
Mature leaf color	Green	Light green	Dark green

**Table 2 biology-14-01638-t002:** Comparative analysis of free amino acid differences in *Macadamia integrifolia* leaves.

Amino-Acid Types (μg/g)	A4	A16	A203
Asp	259.91 ± 2.65 c	339.12 ± 3.55 a	277.46 ± 5.32 b
Thr	0.64 ± 0.16 a	0.45 ± 0.12 ab	0.11 ± 0.05 b
Ser	549.26 ± 9.02 b	1175.76 ± 10.54 a	450.28 ± 7.51 c
Glu	228.17 ± 5.95 c	350.2 ± 4.87 a	289.65 ± 6.78 b
Gly	4.04 ± 0.29 b	5.74 ± 0.15 a	5.14 ± 0.26 a
Ala	21.67 ± 0.35 a	10.59 ± 0.38 b	6.7 ± 0.34 c
Cys	405.56 ± 12.77 b	496.52 ± 13.58 a	504.69 ± 8.48 a
Val	26.56 ± 1.17 b	42.57 ± 0.74 a	21.53 ± 0.98 c
Met	1.41 ± 0.09 a	1.67 ± 0.07 a	1.66 ± 0.11 a
Ile	7.27 ± 0.64 b	9.76 ± 0.44 a	8.31 ± 0.67 ab
Leu	11.11 ± 0.04 c	12.32 ± 0.05 b	13.79 ± 0.15 a
Tyr	8.48 ± 0.84 b	13.96 ± 0.7 a	7.26 ± 0.78 b
Phe	60.1 ± 1.45 a	52.89 ± 1.35 b	2.65 ± 0.16 c
His	35.32 ± 3.62 a	25.96 ± 0.78 b	5.93 ± 0.37 c
Lys	28.73 ± 0.67 a	29.64 ± 0.83 a	23.84 ± 0.77 b
Arg	1128.64 ± 29.83 b	1269.07 ± 15.3 a	536.09 ± 15.1 c
Pro	531.64 ± 15.29 a	372.12 ± 4.62 b	336.59 ± 6.25 c
Total amino acids	3308.51	4208.31	2491.67

Note: Different lowercase letters in the same column indicate significant differences (*p* < 0.05). Data are presented as mean ± standard deviation (μg/g fresh weight).

**Table 3 biology-14-01638-t003:** Comparative analysis of free amino acid differences in *Macadamia* nut kernels.

Amino-Acid Types (μg/g)	A4	A16	A203
Asp □ ○	501.63 ± 10.56 b	522.17 ± 2.39 b	701.4 ± 33.94 a
Thr ◊ *	75.13 ± 3.82 b	121.4 ± 1.51 a	54.08 ± 3.87 c
Ser ◊	324.78 ± 9.24 a	353.76 ± 4.18 a	135.96 ± 15.11 b
Glu □ ○	362.9 ± 31.69 b	265.32 ± 12.48 c	625.73 ± 17.74 a
Gly □ ◊	19.03 ± 0.97 b	31.28 ± 1.65 a	11.28 ± 1.21 c
Ala □ ◊	206.67 ± 11.05 b	93.63 ± 6.01 c	359.14 ± 17.08 a
Cys	4.24 ± 0.18 a	1.08 ± 0.04 c	2.61 ± 0.11 b
Val Δ *	92.7 ± 2.31 a	27.01 ± 1.96 c	58.05 ± 3.76 b
Met Δ *	44.08 ± 1.65 a	14.95 ± 2.03 b	11.04 ± 1.07 b
Ile □ Δ *	38.49 ± 0.55 a	9.95 ± 1.54 c	16.25 ± 1.12 b
Leu □ Δ *	23.23 ± 0.53 a	5.2 ± 0.51 b	22.64 ± 1.18 a
Tyr ⌂	34.02 ± 2.98 b	94.12 ± 1.8 a	14.25 ± 1.28 c
Phe □ Δ ⌂ *	75 ± 4.71 c	169.31 ± 4.5 a	126.82 ± 4.15 b
His ◊ Δ	11.22 ± 0.65 b	43.57 ± 0.34 a	8.84 ± 1.07 b
Lys □ ○ *	80.19 ± 3.9 b	212.64 ± 8.91 a	79.01 ± 4.46 b
Arg □ Δ	42.2 ± 4.06 b	305.51 ± 3.46 a	27.61 ± 2.43 c
Pro ◊	24.76 ± 2.62 b	436.16 ± 6.83 a	24.07 ± 5.47 b
Total amino acids	1960.28	2707.04	2278.77

Note: Different lowercase letters in the same row indicate significant differences (*p* < 0.05). Symbols denote functional classifications: □ pharmacological amino acids, ○ umami amino acids, ◊ sweet amino acids, Δ bitter amino acids, * essential amino acids for humans, and ⌂ aromatic amino acids. Data are presented as mean ± standard deviation (μg/g fresh weight).

## Data Availability

Data will be made available on request.

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
