# Peer review of "Screening of Macadamia integrifolia Varieties Based on the Comparison of Seedling Adaptability and Quality Differences"

_biology, 2025, doi:10.3390/biology14121638_

Round 1
Reviewer 1 Report
Comments and Suggestions for Authors
The research topic is of considerable interest. This study presents a comparative analysis of three macadamia varieties (A4, A16, A203), evaluating their performance based on the physiological characteristics of their leaves and mature fruits.
The foliar analysis encompasses morphological parameters, photosynthetic performance, enzymatic activities, and amino acid content. Concurrently, the mature fruit analysis includes morphological assessment, mineral composition, amino acid profile, and the quantification of phenolic compounds in the pericarp.
The existing literature establishes that macadamia nuts are primarily valued for their high content of oil, protein, and sugars. This study provides valuable insights that will assist future researchers, agricultural practitioners, and policymakers in selecting superior macadamia varieties yielding high-quality nuts suitable for commercial markets.
However, the inclusion of additional information would strengthen the manuscript.
Introduction:
The introduction is well-composed. It would be strengthened by a brief justification -one or two sentences- elucidating the economic significance of the macadamia tree and the rationale for its selection.
Materials and Methods:
This section provides a clear description of the plant material and analytical methodologies. To enhance the work, the inclusion of supplementary data is recommended.
Notably, pedo-climatic data (environmental parameters: soil and climat) for the study area are absent; this information is essential for a robust contextualization and discussion of the experimental results.
On the other hand, I would like to know more about the pollination type of these three varieties, as it strongly influences fruit quality.
Results:
The results are clearly and effectively presented, with data well-supported by tables and figures.
Discussion:
The discussion is well-articulated. A valuable addition would be a segment addressing the pollination method of these varieties and its potential influence on final fruit quality.
Conclusion:
The conclusion should be comprehensively articulated to effectively synthesize the study's findings.
Author Response
The research topic is of considerable interest. This study presents a comparative analysis of three macadamia varieties (A4, A16, A203), evaluating their performance based on the physiological characteristics of their leaves and mature fruits.
The foliar analysis encompasses morphological parameters, photosynthetic performance, enzymatic activities, and amino acid content. Concurrently, the mature fruit analysis includes morphological assessment, mineral composition, amino acid profile, and the quantification of phenolic compounds in the pericarp.
The existing literature establishes that macadamia nuts are primarily valued for their high content of oil, protein, and sugars. This study provides valuable insights that will assist future researchers, agricultural practitioners, and policymakers in selecting superior macadamia varieties yielding high-quality nuts suitable for commercial markets.
However, the inclusion of additional information would strengthen the manuscript.
Introduction:
The introduction is well-composed. It would be strengthened by a brief justification -one or two sentences- elucidating the economic significance of the macadamia tree and the rationale for its selection.
Answer: We thank the reviewers for their valuable comments. In response, we have added a concise description in the Introduction on the economic relevance of macadamia nuts and the basis for their selection, thereby providing better contextual justification for the study. The revised text is provided below:
“Macadamia (Macadamia spp.), an evergreen tree fruit crop belonging to the Proteaceae family and the genus Macadamia F. Muell., is native to the subtropical rainforests of southeastern Queensland, Australia [1]. ... . As a high-value cash crop, macadamia is increasingly cultivated in tropical and subtropical regions for its premium edible nuts, which command strong market prices and contribute significantly to local economies. In recent years, with the rapid growth of the health food market, macadamia has found increasingly wide applications in the food, health product, and cosmetics industries, demonstrating promising market prospects [4,5]. ... . Therefore, selecting superior varieties with enhanced adaptability and quality traits is crucial for improving production efficiency and industrial competitiveness. Furthermore, bioactive components like phenolics and flavonoids, ... ” (lines 65-78)
Materials and Methods:
This section provides a clear description of the plant material and analytical methodologies. To enhance the work, the inclusion of supplementary data is recommended.
Answer: We thank the reviewers for their positive feedback on the Materials and Methods section and for their valuable suggestions. We understand your request for more comprehensive data support in the methodology. Accordingly, we have added a new Section “Analysis of amino acid classification” to the "Materials and Methods" to detail the analytical methods for the supplementary data.
“2.9. Analysis of amino acid classification
The classification of free amino acids into functional categories (including umami, sweet, bitter, aromatic, medicinal, and essential amino acids) was performed based on established taste and functional attributes reported in the literature [9,10]. Principal component analysis (PCA) was conducted using the OmicShare Tools platform to visualize the overall differences in fruit quality traits among varieties. ” (lines 301-306)
Notably, pedo-climatic data (environmental parameters: soil and climat) for the study area are absent; this information is essential for a robust contextualization and discussion of the experimental results.
Answer: We thank the reviewer for raising this important point. We fully agree that soil and climate data are crucial for a comprehensive understanding of plant physiological responses and fruit quality development. The primary objective of this study was to compare the intrinsic physiological and quality differences among varieties under uniform field management conditions. Therefore, the experimental design focused on controlling cultivation practices to highlight varietal characteristics. We acknowledge that the lack of systematic environmental data is a limitation. In response to this concern, we have supplemented the "2.1. Plant Materials" section in Materials and Methods with the geographic location and altitude of the sampling site, which are key factors determining the local macro-climatic conditions. Additionally, we have added a statement at the beginning of the Discussion to explicitly acknowledge this limitation and identify it as an important direction for future research.
“The fresh, green-in-husk nuts were harvested from the Manxieba Nut Base (altitude ~1269 m, 22°N, 100°E) of Pu'er Yunguo Agricultural Technology Co., Ltd. in Simao District, Pu'er City. This region is characterized by a subtropical highland climate, with the specific soil and climatic conditions representing a typical production area for macadamia in Yunnan. Nuts were collected from four-year-old plants (fertilized 4-5 times per year) located on sunny slopes at consistent altitudes. ... ” (lines 148-151)
“It is important to note that this study was conducted under the specific pedo-climatic conditions of the experimental site in Pu'er, Yunnan. While this provides a unified context for comparing the relative performance of the three varieties, the absence of detailed soil profiles and continuous microclimatic data limits the extrapolation of the absolute values of these physiological and quality traits to other environments. Future studies incorporating multi-environment trials will be valuable to dissect the genotype-by-environment interactions. Nonetheless, the observed differences robustly reflect the distinct genetic backgrounds and physiological strategies of the varieties under the same managed conditions.” (lines 607-615)
On the other hand, I would like to know more about the pollination type of these three varieties, as it strongly influences fruit quality.
Answer: We thank the reviewer for raising this important question. We agree that the pollination method can significantly influence fruit quality. In response to this comment, we have supplemented the description of each cultivar in the "2.1. Plant Materials" section of Materials and Methods with their known pollination characteristics. This additional information indicates that the selected cultivars in this study exhibit differences in pollination biology, thereby providing a partial biological context for the observed variations in fruit quality.
“Their specific characteristics are as follows: (1) A4: This cultivar exhibits early flowering, spiny leaf margins, and a high degree of self-sterility, with relatively poor compatibility with common rootstocks. It is resistant to husk spot disease, exhibits strong cold and wind tolerance, but is susceptible to high-temperature stress. A4 is suitable for processing and produces high-quality kernels. Its cultivation area in Yunnan Province is expanding. (2) A16: Possessing some hybrid characteristics, A16 grows vigorously with moderately dense branches. It is high-yielding, producing medium-sized, uniform nuts with high kernel recovery, high percentages of first-grade and whole kernels, and excellent kernel quality. Its pollination biology is not well-documented, but it is typically cultivated in mixed orchards to ensure cross-pollination. It demonstrates strong wind, drought, and cold tolerance. Its planting area in Yunnan is also increasing. (3) A203: This cultivar forms a natural shape with natural branching. It flowers and fruits early, bearing nuts in clusters. The fruits are relatively large, and the cultivar is high-yielding with a relatively higher fruit set rate compared to other varieties. It is considered to have a mixed pollination system but benefits from cross-pollination.” (lines 123-136)
“Furthermore, as noted in the plant materials, varietal differences in pollination type (e.g., the high self-sterility of A4) may have also contributed to the observed variations in fruit set and kernel quality parameters. While the experimental orchard provided opportunities for cross-pollination, the inherent pollination efficiency and compatibility of each variety represent an intrinsic trait that influences yield and quality formation, and should be considered in future breeding and orchard management strategies.” (lines 881-886)
Results:
The results are clearly and effectively presented, with data well-supported by tables and figures.
Answer: We thank the reviewer for their positive assessment of the presentation of our results. We are pleased that you found the data and figures to be clear and to effectively support the study's conclusions, which we attribute to the rigorous standards consistently maintained throughout our experimental design and data analysis. We remain committed to upholding this scientific approach in our future work.
Discussion:
The discussion is well-articulated. A valuable addition would be a segment addressing the pollination method of these varieties and its potential influence on final fruit quality.
Answer: We thank the reviewer for their positive feedback on the Discussion and for this important constructive suggestion. We agree that exploring pollination, as a key biological process influencing fruit development, will contribute to a more comprehensive interpretation of the mechanisms underlying fruit quality formation. Relevant content has been incorporated into the Discussion section.
“Beyond nutritional composition, pollination method represents a fundamental biological factor influencing fruit development and quality. As documented in the plant materials, the three varieties exhibit distinct pollination characteristics: A4 is highly self-sterile, while both A16 and A203 show a tendency for cross-pollination. These differential pollination behaviors may contribute to the observed variations in fruit set efficiency, kernel development, and ultimately, the accumulation of specific quality components [14]. In macadamia, cross-pollination has been associated with improved kernel recovery rate and more uniform nut development, which aligns with the superior fruit weight and morphological consistency observed in A16 and A203 compared to the highly self-sterile A4. Future studies specifically designed to dissect the pollination regime's effect on metabolic profiles (e.g., amino acids, lipids) of these varieties would provide deeper insights into the interaction between reproductive biology and nut quality formation.” (lines 868-880)
Conclusion:
The conclusion should be comprehensively articulated to effectively synthesize the study's findings.
Answer: We thank the reviewer for their valuable suggestions on the Conclusion section. We have thoroughly revised this section in accordance with the comments to provide a more systematic synthesis of the key findings and a clearer articulation of the theoretical and practical implications of our study.
“This study systematically compared the differences ... . The results demonstrate that each variety possesses a unique combination of traits, forming specific adaptation and quality profiles: A4 exhibited outstanding performance in chlorophyll content, antioxidant enzyme activity, and husk phenolic compound accumulation, which underscores its strong stress resistance potential and high value for by-product utilization. A16 showed significant advantages in photosynthetic efficiency, water use efficiency, and total amino acid content in kernels, particularly its richness in flavor amino acids, making it highly suitable for fresh consumption and high-value product development. A203 excelled in light use efficiency, accumulation of mineral elements (such as K, Mg, Mn), and the proportion of medicinal amino acids, demonstrating superior potential for nutritional functionality.
Through multi-dimensional correlation analysis and principal component analysis, this study further elucidated the intrinsic relationships between leaf phenotype and physiological status, as well as the complex regulatory network between fruit morphology and intrinsic quality. More importantly, we established a comprehensive multi-trait evaluation system encompassing "morphology—photosynthesis— antioxidant activity—amino acids—quality," which provides a scientific framework for variety assessment. The identified variety-specific trait profiles provide clear targets for future breeding programs, enabling the selection of parental lines for developing new cultivars with desired stress resistance, flavor, or nutritional traits. Furthermore, the physiological adaptation strategies revealed at the seedling stage offer valuable criteria for designing regional adaptation experiments to match specific varieties with optimal growing environments.
In practical applications, these findings offer crucial guidance for targeted variety selection and cultivation management: A4 is recommended for cultivation under adverse conditions or for by-product utilization; A16 is ideal for fresh consumption and high-nutritional-value product development; while A203 shows great promise for nutritionally fortified products. The comprehensive evaluation system established in this study not only provides a theoretical basis for macadamia variety breeding and whole-industry-chain development but also lays a scientific foundation for promoting sustainable development and value-added utilization of the nut industry.” (lines 980-1009)
Reviewer 2 Report
Comments and Suggestions for Authors
Reviewer Report
The paper adopts a comparative analysis of three varieties of Macadamia integrifolia in the form of seedling adaptability, physiological parameters, antioxidant activity, and fruit quality attributes. The study fills an appropriate research gap in the breeding of macadamia - there is a deficiency in thorough estimation of the seedling stage, and the study is informative in terms of improvement of the varietal and industrial use. The paper is clearly structured and has a sound methodology, through the massive gathering of data and evident statistical interpretation. Nevertheless, the manuscript needs some improvements to make it easier to read and understand and explore the mechanisms behind it. The use of English expression, structures of figures and organization of a discussion should also be refined a bit.
Major Comments
- It is a good effort on the part of the study to give valuable information on the incorporation of physiological and quality parameters in varietal screening. However, the introduction can state more clearly what makes the given study stand out of previous macadamia variety assessments
- The choice of the three varieties is valid, but the conditions associated with the environment in the cultivation process ought to be mentioned better to facilitate reproducibility.
- The figures and tables are properly arranged, although some of them need a better labeling.
- It has been successful in connecting physiological performance with adaptability but it can be improved with correlations between biochemical characteristics and ecological consequences.
- The ROS signaling and metabolic regulation should be viewed through the mechanism of the differential antioxidant enzyme activity (CAT, SOD, POD) of the varieties.
- The possible industrial applications of particular amino acid and phenolic profiles may be addressed more thoroughly.
- The paper is generally understandable, but there are too long and redundant sentences. Suggestions of minor grammatical revisions.
- The italics should always be used when giving scientific names.
Minor Comments
- Abstract: is well organized yet a bit long. The concluding sentence might briefly highlight the usefulness of the study conducted in the breeding and industry of macadamia.
- In conclusion include one or two sentences on how these results may be used in future breeding or regional adaptation experiments.
Author Response
The paper adopts a comparative analysis of three varieties of Macadamia integrifolia in the form of seedling adaptability, physiological parameters, antioxidant activity, and fruit quality attributes. The study fills an appropriate research gap in the breeding of macadamia - there is a deficiency in thorough estimation of the seedling stage, and the study is informative in terms of improvement of the varietal and industrial use. The paper is clearly structured and has a sound methodology, through the massive gathering of data and evident statistical interpretation. Nevertheless, the manuscript needs some improvements to make it easier to read and understand and explore the mechanisms behind it. The use of English expression, structures of figures and organization of a discussion should also be refined a bit.
Major Comments
- It is a good effort on the part of the study to give valuable information on the incorporation of physiological and quality parameters in varietal screening. However, the introduction can state more clearly what makes the given study stand out of previous macadamia variety assessments.
Answer: We sincerely thank the reviewer for the positive feedback and valuable suggestion. As recommended, we have revised the Introduction section to more clearly articulate the unique contributions of our study compared to previous macadamia variety assessments. Specifically, we have emphasized the establishment of a comprehensive, multi-trait evaluation system that integrates seedling physiological adaptability with fruit quality attributes—an approach not previously reported in macadamia research.
“Although previous research has conducted regional adaptability evaluations for introduced macadamia cultivars, systematic investigation into physiological differences at the seedling stage – particularly regarding leaf free amino acids and comprehensive multi-indicator assessments – remains limited. More importantly, existing studies often evaluate seedling adaptability and fruit quality in isolation, lacking an integrated framework that links early physiological performance with final product attributes. To address this gap, we established a comprehensive multi-trait evaluation system encompassing "morphology– photosynthesis–antioxidant activity–amino acids–quality," which simultaneously assesses seedling adaptability and mature fruit quality. Therefore, ...” (lines 104-109)
- The choice of the three varieties is valid, but the conditions associated with the environment in the cultivation process ought to be mentioned better to facilitate reproducibility.
Answer: We thank the reviewer for this valuable suggestion to improve the description of the cultivation conditions. We have revised the Plant Materials section to provide a clearer and more structured description of the environmental context and management practices, which will facilitate the reproducibility of the study.
“They were cultivated in the open-field macadamia nursery of the College of Tropical Crops, Yunnan Agricultural University. The nursery site is located in a region with a typical subtropical highland monsoon climate. The seedlings were grown under uniform management practices, including consistent irrigation and fertilization regimes, reflecting common commercial nursery conditions in the region. Mature leaves were collected from these seedlings for physiological indicator measurements related to adaptability. The fresh, green-in-husk nuts were harvested from the Manxieba Nut Base (altitude ~1269 m, 22°N, 100°E) of Pu'er Yunguo Agricultural Technology Co., Ltd. in Simao District, Pu'er City. This region is characterized by a subtropical highland climate, with the specific soil and climatic conditions representing a typical production area for macadamia in Yunnan. Nuts were collected from ...” (lines 142-151)
- The figures and tables are properly arranged, although some of them need a better labeling.
Answer: We thank the reviewer for the positive feedback on the figures and tables and for the suggestion regarding labeling. We have carefully reviewed all figures and tables and implemented improvements to ensure all labels, legends, and annotations are clear, consistent, and sufficiently detailed. The key modifications are highlighted in the revised captions and table notes below.
“Figure 2. Analysis of photosynthetic efficiency parameters of Macadamia integrifolia leaves. (A) SPAD value, (B) Total chlorophyll content, (C) Chlorophyll-a content, (D) Chlorophyll-b content, (E) Net Photosynthetic rate (Pn), (F) Transpiration rate (Tr), (G) Stomatal conductance (Cleaf), (H) Intercellular CO₂ concentration (Ci), (I) Light use efficiency (LUE), (J) Water efficiency (WUE), (K) Leaf temperature (Tleaf), (L) Stomatal impedance (Si). The results are presented as the means ± SDs (n=3). Different letters indicate significant differences (P < 0.05 according to Tukey’s test).” (lines 372-375)
“Figure 5. Analysis of phenotypic differences in Macadamia nuts. (A) Comparison of fresh fruits of cultivars A4, A16, and A203, (B) Fresh fruit diameter (horizontal diameter), (C) Fresh fruit longitudinal diameter, (D) Fresh fruit weight, (E) Comparison of fresh pericarps of cultivars A4, A16, and A203, (F) Fresh shelled fruit weight, (G) Fresh husk of cultivars A4, A16, and A203, ... .” (lines 472-474)
“Note: Different lowercase letters in the same column indicate significant differences (P < 0.05). Data are presented as mean ± standard deviation (μg/g fresh weight).” (lines 425)
“Note: Different lowercase letters in the same row indicate significant differences (P < 0.05). Symbols denote functional classifications: □ pharmacological amino acids, ○ umami amino acids, ◊ sweet amino acids, Δ bitter amino acids, * essential amino acids for humans, and ⌂ aromatic amino acids. Data are presented as mean ± standard deviation (μg/g fresh weight).” (lines 510-513)
“Figure 8. Correlation analysis of fruit phenotypes and quality indicators of Macadamia nuts. (A) Principal Component Analysis(PCA) score plot showing the separation of the three varieties (A4, A16, A203) based on combined fruit phenotype and quality data. (B) Heatmap of intergroup correlations between fruit phenotypic indicators (columns) and kernel quality indicators (rows). (C) Heatmap of intragroup correlations among kernel quality indicators (mineral elements, amino acids, phenolic compounds). In panels (B) and (C), the phenolic compounds are labeled as Phenol-1 to Phenol-4, corresponding to p-hydroxybenzyl alcohol, 3,4-dihydroxybenzoic acid, p-hydroxybenzoic acid, and p-hydroxybenzaldehyde, respectively. Dark blue represents ... ” (lines 596-602)
- It has been successful in connecting physiological performance with adaptability but it can be improved with correlations between biochemical characteristics and ecological consequences.
Answer: We sincerely thank the reviewer for this insightful suggestion. To better connect the biochemical characteristics with their broader ecological implications, we have enhanced the Discussion. Specifically, we have added a new paragraph that explicitly discusses how the observed biochemical traits (e.g., antioxidant enzymes, amino acids) relate to ecological strategies and consequences for the varieties.
“The distinct biochemical profiles observed among the varieties have direct ecological consequences for their adaptability and potential niche differentiation. The superior antioxidant capacity (high CAT, low MDA) in A4 signifies a robust constitutive defense system, an advantageous strategy for pre-adaptation to environments prone to frequent abiotic stresses such as high light, temperature fluctuations, or poor soil conditions. This biochemical investment in stress mitigation likely underlies its reputation for strong cold and wind tolerance. In contrast, A203's strategy of high light use efficiency (LUE) coupled with efficient O²⁻ scavenging (high SOD, low O²⁻ content) suggests an adaptation optimized for maximizing carbon gain under moderate stress levels, potentially favoring competitive growth in stable but light-limited environments. Furthermore, the high nitrogen metabolic activity (reflected by high total free amino acid content) in A16 aligns with a resource- acquisitive strategy, supporting rapid growth and high yield under favorable conditions with ample nutrient and water availability. Thus, the interspecific variation in key biochemical traits—antioxidant systems, photosynthetic efficiency, and nitrogen metabolism—not only explains the differential physiological performance but also predicts their ecological success and suitable habitats, providing a mechanistic basis for strategic cultivar deployment in diverse agro-ecological zones.” (lines 742-757)
- The ROS signaling and metabolic regulation should be viewed through the mechanism of the differential antioxidant enzyme activity (CAT, SOD, POD) of the varieties.
Answer: We thank the reviewer for this insightful suggestion to deepen the mechanistic interpretation of our findings regarding ROS signaling and metabolic regulation. We have revised the Discussion (specifically, a portion of section 4.2) to explicitly frame the observed differences in ROS homeostasis and amino acid profiles through the lens of the differential antioxidant enzyme activities (CAT, SOD, POD) among the varieties.
“... The distinct configurations of the antioxidant enzyme system (CAT, SOD, POD) provide a mechanistic basis for interpreting the observed ROS signaling and associated metabolic regulation across the varieties. In A4, the significantly highest CAT activity, coupled with the lowest MDA content, indicates a highly efficient mechanism for H₂O₂ detoxification. This robust CAT-mediated clearance likely maintains H₂O₂ below damaging thresholds, yet potentially permits its function as a signaling molecule within a sub-toxic range, fostering a cellular environment conducive to the observed accumulation of osmoprotectants like Pro and Ala. This enzymatic strategy underpins A4's constitutive stress tolerance. Conversely, A203's profile was dominated by the highest SOD activity, facilitating the rapid dismutation of superoxide (O²⁻), which aligns with its lowest O²⁻ content. This prioritization of primary ROS conversion at the source minimizes the generation of more reactive secondary species and reduces the metabolic load on downstream scavengers like POD (which was lowest in A203). This efficient O²⁻ management likely contributes to membrane stability (low MDA) and supports high light-use efficiency by mitigating photo-oxidative damage. The coordinated yet distinct activities of these antioxidant enzymes in each variety not only define their capacity to scavenge specific ROS but also shape the spatiotemporal dynamics of the ROS landscape (e.g., H₂O₂/O²⁻ ratio), which are known to differentially modulate transcription factors and calcium signaling, thereby influencing the expression of metabolic genes. This could explain the varietal divergence in nitrogen assimilation and amino acid partitioning, such as the high nitrogen flux towards Asp, Ser, and Glu in A16 versus the preferential accumulation of specific osmolytes in A4. Thus, the genotype-specific blueprint of the antioxidant enzyme network is a fundamental regulator that connects ROS metabolism to broader metabolic reprogramming and stress adaptation strategies.” (lines 670-692)
- The possible industrial applications of particular amino acid and phenolic profiles may be addressed more thoroughly.
Answer: We thank the reviewer for this valuable suggestion to elaborate on the industrial potential of the identified amino acid and phenolic profiles. We have revised the Discussion (specifically, subsections 4.5 and 4.6) to provide a more thorough and targeted analysis of the possible industrial applications, linking the specific biochemical traits of each variety to potential high-value product development.
“The distinct amino acid profiles of the kernels reveal clear directions for targeted industrial applications. The high abundance of umami and sweet amino acids (e.g., Glu, Asp, Ala, Gly, Ser) in A16 kernels directly contributes to a superior, inherent savory and sweet taste profile. This makes A16 particularly suitable for high-value markets such as premium fresh consumption, gourmet snacks, and as a natural flavor enhancer in the culinary and food manufacturing industries, potentially reducing the need for added monosodium glutamate or other synthetic flavorings. In contrast, A203, with its exceptionally high proportion of medicinal amino acids (e.g., specific profiles of Arg, Leu, Ile), holds significant promise for the nutraceutical and functional food sectors. Kernels or extracts from A203 could be developed into targeted nutritional supplements aimed at supporting specific physiological functions, such as immune modulation or metabolic health. Furthermore, the unique and balanced amino acid composition across all varieties underscores their high value as a source of plant-based proteins for the health food market.” (lines 854-867)
“In this study, the accumulation of phenolic substances like 3,4-dihydroxybenzoic acid and p-hydroxybenzoic acid in the husks of variety A4 was significantly higher than in A16 and A203, indicating high antioxidant potential (Figure 7). The distinct amino acid profiles of the kernels reveal clear directions for targeted industrial applications. The high abundance of umami and sweet amino acids (e.g., Glu, Asp, Ala, Gly, Ser) in A16 kernels directly contributes to a superior, inherent savory and sweet taste profile. This makes A16 particularly suitable for high-value markets such as premium fresh consumption, gourmet snacks, and as a natural flavor enhancer in the culinary and food manufacturing industries, potentially reducing the need for added monosodium glutamate or other synthetic flavorings. In contrast, A203, with its exceptionally high proportion of medicinal amino acids (e.g., specific profiles of Arg, Leu, Ile), holds significant promise for the nutraceutical and functional food sectors. Kernels or extracts from A203 could be developed into targeted nutritional supplements aimed at supporting specific physiological functions, such as immune modulation or metabolic health. Furthermore, the unique and balanced amino acid composition across all varieties underscores their high value as a source of plant-based proteins for the health food market.” (lines 903-917)
“The levels of these two indicators are key factors determining final fruit quality. Ultimately, the synergistic consideration of kernel amino acids and husk phenolics enables a whole-industry-chain optimization strategy. For instance, A16 fruits, with their superior kernel flavor, should be directed towards the fresh nut and high-end confectionery markets to maximize economic return. Simultaneously, A4, with its smaller fruit size but phenolics-rich husk, presents a compelling case for integrated processing where the primary product (kernel) and the by-product (husk) are both efficiently utilized. This dual-value approach not only enhances the economic resilience of macadamia cultivation but also aligns with circular bio-economy principles. Our study provides the scientific rationale for such varietal specialization in breeding programs and industrial processing.” (lines 937-947)
- The paper is generally understandable, but there are too long and redundant sentences. Suggestions of minor grammatical revisions.
Answer: We sincerely thank the reviewer for the valuable feedback regarding sentence length and clarity. We agree that concise and clear expression enhances readability. We have carefully reviewed the entire manuscript and revised sentences that were overly long or redundant to improve flow and comprehension, while strictly maintaining all scientific content and accuracy. The changes are focused on simplifying sentence structure, removing unnecessary words, and splitting complex sentences where appropriate.
“... We established a comprehensive evaluation system encompassing "morphology —photosynthesis—antioxidant activity—amino acids— quality," which provides a scientific basis for variety selection, cultivation management, and industrial development of macadamia.” (lines 21-24)
“... However, existing research has predominantly focused on the mature tree stage. Systematic studies on the physiological characteristics during the seedling stage and comprehensive multi-indicator evaluations remain insufficient, limiting improved variety selection and industrial development. This study investigated three macadamia varieties (A4, A16, A203). We systematically measured leaf morphology, photosynthetic parameters, antioxidant enzyme activities, ...” (lines 30-34)
“... Therefore, we utilized cultivars A4, A16, and A203 as plant materials. We measured seedling leaf morphology, photosynthetic parameters, antioxidant enzyme activities, and free amino acid profiles, combined with analyses of kernel nutritional components and husk phenolic substances, to systematically compare seedling adaptability and fruit quality differences among the cultivars. These findings provide theoretical support for elite cultivar selection and industry development.” (lines 109-115)
“... Differences in the correlations among indicators across varieties suggest that varietal characteristics may influence the interrelationships of these physiological parameters, providing clues for studying differences in physiological traits among varieties. These correlations help understand the intrinsic links among various physiological processes in macadamia seedlings, offering theoretical references for variety selection and cultivation management optimization.” (lines 739-741)
- The italics should always be used when giving scientific names.
Answer: We sincerely thank the reviewer for pointing out the need for consistent formatting of scientific names. We have carefully reviewed the entire manuscript and corrected the formatting of all scientific names to ensure they are consistently presented in italics, in accordance with standard academic practice. The changes are listed below.
“The leaf shape index can significantly influence photosynthetic capacity. Research on cassava (Manihot esculenta Crantz) found that changes in leaf morphology lead to ...” (lines 760)
Minor Comments
- Abstract: is well organized yet a bit long. The concluding sentence might briefly highlight the usefulness of the study conducted in the breeding and industry of macadamia.
Answer: We thank the reviewer for the positive feedback on the Abstract's organization and the constructive suggestion to enhance its conciseness and highlight the practical implications. We have revised the abstract by condensing several sentences and adding a concluding sentence that explicitly states the study's usefulness for macadamia breeding and industry.
“Macadamia (Macadamia spp.), as a high-value cash crop, relies on varietal adaptability screening and quality optimization for enhanced industrial benefits. ... Systematic studies on the physiological characteristics during the seedling stage and comprehensive multi-indicator evaluations remain insufficient, limiting improved variety selection and industrial development. This study investigated three macadamia varieties (A4, A16, A203). We systematically measured leaf morphology, photosynthetic parameters, ... . Correlation analysis revealed a complex regulatory network among fruit traits, mineral elements, amino acids, and phenolics. In summary, A4, A16, and A203 possess respective advantages in high stress resistance, superior flavor quality, and high nutritional functionality. This study establishes a comprehensive "morphology–photosynthesis–antioxidant activity–amino acids–quality" evaluation system, providing a scientific basis for targeted breeding and whole-industry-chain development of macadamia.” (lines 30-55)
- In conclusion include one or two sentences on how these results may be used in future breeding or regional adaptation experiments.
Answer: We thank the reviewer for this valuable suggestion. We have revised the Conclusion by adding specific sentences to highlight the potential applications of our findings in future breeding and regional adaptation experiments.
“Through multi-dimensional correlation analysis and principal component analysis, this study further elucidated the intrinsic relationships between leaf phenotype and physiological status, as well as the complex regulatory network between fruit morphology and intrinsic quality. More importantly, we established a comprehensive multi-trait evaluation system encompassing "morphology—photosynthesis— antioxidant activity—amino acids—quality," which provides a scientific framework for variety assessment. The identified variety-specific trait profiles provide clear targets for future breeding programs, enabling the selection of parental lines for developing new cultivars with desired stress resistance, flavor, or nutritional traits. Furthermore, the physiological adaptation strategies revealed at the seedling stage offer valuable criteria for designing regional adaptation experiments to match specific varieties with optimal growing environments.” (lines 990-1001)